# Dopaminergic modulation of the exploration/exploitation trade-off in human decision-making

**Karima Chakroun[1†], David Mathar[2†*], Antonius Wiehler[1,3], Florian Ganzer[4], Jan Peters[1,2]**

[1]Department of Systems Neuroscience, University Medical Center Hamburg-Eppendorf, Hamburg, Germany; [2]Department of Psychology, Biological Psychology, University of Cologne, Cologne, Germany; [3]Institut du Cerveau et de la Moelle épinière - ICM, Centre de NeuroImagerie de Recherche - CENIR, Sorbonne Universités, Groupe Hospitalier Pitié-Salpêtrière, Paris, France; [4]German Center for Addiction Research in Childhood and Adolescence, University Medical Center Hamburg-Eppendorf, Hamburg, Germany

**Abstract** Involvement of dopamine in regulating exploration during decision-making has long been hypothesized, but direct causal evidence in humans is still lacking. Here, we use a combination of computational modeling, pharmacological intervention and functional magnetic resonance imaging to address this issue. Thirty-one healthy male participants performed a restless four-armed bandit task in a within-subjects design under three drug conditions: 150 mg of the dopamine precursor L-dopa, 2 mg of the D2 receptor antagonist haloperidol, and placebo. Choices were best explained by an extension of an established Bayesian learning model accounting for perseveration, directed exploration and random exploration. Modeling revealed attenuated directed exploration under L-dopa, while neural signatures of exploration, exploitation and prediction error were unaffected. Instead, L-dopa attenuated neural representations of overall uncertainty in insula and dorsal anterior cingulate cortex. Our results highlight the computational role of these regions in exploration and suggest that dopamine modulates how this circuit tracks accumulating uncertainty during decision-making.

**\*For correspondence:**
dmathar@uni-koeln.de

[†]These authors contributed equally to this work

**Competing interests:** The authors declare that no competing interests exist.

## Introduction

A central aspect of a broad spectrum of decision problems is the weighting of when to exploit, that is to choose a familiar option with a well-known reward value, and when to explore, that is to try an alternative option with an uncertain but potentially higher payoff. This decision dilemma is commonly known as the 'exploration/exploitation trade-off' (*Cohen et al., 2007*; *Addicott et al., 2017*). Striking a balance between exploration and exploitation is essential for maximizing rewards and minimizing costs in the long term (*Addicott et al., 2017*). Too much exploitation prevents an agent from gathering new information in a volatile environment, and fosters inflexibility and habit formation. Too much exploration, on the other hand, may lead to inefficient and inconsistent decision-making, thereby reducing long-term payoffs (*Beeler et al., 2014*; *Addicott et al., 2017*). Despite the high relevance of the explore/exploit trade-off for optimal decision-making, research is only beginning to unravel the mechanisms through which animals and humans solve this dilemma.

Several tasks have been developed to test explore/exploit behavior in both animals and humans. The most widely used paradigm in both human (Daw) and primate work (*Costa et al., 2019*) is the multi-armed bandit task (*Robbins, 1952*; *Gittins and Jones, 1974*). It mirrors a casino's slot-machine with multiple arms. Several implementations exist that differ according to the number of arms and

their underlying reward structure. The restless bandit paradigm uses continuous, slowly drifting rewards for each bandit that encourage participants to strike a balance between exploiting the currently best option and exploring alternative bandits to keep track of their evolving rewards (*Daw et al., 2006*; *Addicott et al., 2017*). Other prominent paradigms include the (patch) foraging task (*Cook et al., 2013*; *Addicott et al., 2014*; Constantino & Daw, 2015) that mirrors exploration and exploitation of food sources in a more naturalistic setting, and the horizon task (*Wilson et al., 2014*) that examines exploration in series of discrete games. These paradigms offer different approaches to measure explore/exploit behavior and may be used to address different research questions. Computational models have served as an elegant tool for modeling behavior on these tasks, yielding insights into latent cognitive processes, and inter individual differences (*Sutton and Barto, 1998*; *Daw et al., 2006*; *Gershman, 2018*).

These computational models have at least two components: a learning rule and a choice rule. The learning rule describes how subjective value estimates of an option's mean outcome are updated for each choice option based on experience, for example via the classical 'Delta rule' (*Rescorla and Wagner, 1972*) from reinforcement learning theory (*Sutton and Barto, 1998*). Work on explore/exploit behavior has also to utilized a 'Bayesian learner' model that relies on a Kalman filter model that simultaneously tracks estimates of outcome mean and uncertainty (e.g. *Daw et al., 2006*; *Speekenbrink and Konstantinidis, 2015*), and updates values based on an uncertainty-dependent delta rule. The choice rule then accounts for how learned values give rise to choices. Here, exploration can be due to at least two mechanisms. First, exploration could result from a probabilistic selection of sub-optimal options as in ε-greedy or softmax choice rules (*Sutton and Barto, 1998*), henceforth referred to as 'random exploration' (*Daw et al., 2006*; *Speekenbrink and Konstantinidis, 2015*). Recently, *Gershman, 2018* reported evidence for random exploration to depend on the summed uncertainty over all choice options, in line with 'Thompson sampling' (*Thompson, 1933*). In contrast, exploration could also be based on the degree of uncertainty associated with a single option (*Daw et al., 2006*; *Wilson et al., 2014*), such that highly uncertain options have a higher probability to be strategically explored by an agent, henceforth referred to as 'directed exploration'. However, estimation of exploration/exploitation behavior might be partially confounded by perseveration (i.e. repeating previous choices irrespective of value or uncertainty), a factor that has not been incorporated in previous models (*Badre et al., 2012*; *Speekenbrink and Konstantinidis, 2015*).

Dopamine (DA) neurotransmission is thought to play a central role in the explore/exploit trade-off. Striatal phasic DA release is tightly linked to reward learning based on reward prediction errors (RPEs) (*Steinberg et al., 2013*) that reflect differences between experienced and expected outcomes, and serve as a 'teaching signal' that update value predictions (*Schultz et al., 1997*; *Schultz, 2016*; *Tsai et al., 2009*; *Glimcher, 2011*; *Chang et al., 2018*). Exploitation has been linked to polymorphisms in genes controlling striatal DA signaling, namely the DRD2 gene (*Frank et al., 2009*) predictive of striatal D2 receptor availability (*Hirvonen et al., 2004*), and the DARPP-32 gene involved in striatal D1 receptor-mediated synaptic plasticity and reward learning (e.g. *Calabresi et al., 2000*; *Stipanovich et al., 2008*). Variation in the slower tonic DA signal might also contribute to an adaptive regulation of exploration/exploitation. *Beeler et al., 2010* found that dopamine-transporter (DAT) knockdown mice that are characterized by increased striatal levels of tonic DA (*Zhuang et al., 2001*) showed higher random exploration compared to wild-type controls. In addition to striatal DA, prefrontal DA might also be involved in explore/exploit behavior. In humans, both directed and random exploration have been associated with variations in the catechol-O-methyltransferase (COMT) gene (*Kayser et al., 2015*; *Gershman and Tzovaras, 2018*) that modulates prefrontal DAergic tone (*Meyer-Lindenberg et al., 2005*). Participants with putatively higher prefrontal DA tone had highest levels of exploration.

These findings regarding the roles of striatal and frontal DA in exploration resonate with cognitive neuroscience studies suggesting that exploration and exploitation rely on distinct neural systems. *Daw et al., 2006* showed that frontopolar cortex (FPC) is activated during exploratory choices, possibly facilitating behavioral switching between an exploitative and exploratory mode by overriding value-driven choice tendencies (*Daw et al., 2006*; *Badre et al., 2012*; *Addicott et al., 2014*; *Mansouri et al., 2017*). In line with this idea, up- and down-regulation of FPC excitability via transcranial direct current stimulation (TDCS) increases and decreases exploration during reward-based learning (*Raja Beharelle et al., 2015*). Anterior cingulate cortex (ACC) and anterior insula (AI), have

also been implicated in exploration (*Addicott et al., 2014*; *Laureiro-Martínez et al., 2014*; *Laureiro-Martínez et al., 2015*; *Blanchard and Gershman, 2018*), although their precise computational role remains elusive (*Blanchard and Gershman, 2018*). Both regions may trigger attentional reallocation to salient choice options in the light of increasing uncertainty (*Laureiro-Martínez et al., 2015*). Exploitation, on the other hand, is thought to be predominantly supported by structures within a 'valuation' network including ventromedial prefrontal cortex (vmPFC), orbitofrontal cortex (OFC), ventral striatum and hippocampus (*Daw et al., 2006*; *Bartra et al., 2013*; *Clithero and Rangel, 2014*; *Laureiro-Martínez et al., 2014*; *Laureiro-Martínez et al., 2015*).

Despite these advances, evidence for a causal link between DA transmission, exploration and the underlying neural mechanisms in humans is still lacking. To address this issue, we combined computational modeling and functional magnetic resonance imaging (fMRI) with a pharmacological intervention in a double-blind, counterbalanced, placebo-controlled within-subjects study. Participants performed a restless four-armed bandit task (*Daw et al., 2006*) during fMRI under three drug conditions: the DA precursor L-dopa (150 mg), the DRD2 antagonist haloperidol (2 mg), and placebo. While L-dopa is thought to stimulate DA transmission by providing increased substrate for DA synthesis, haloperidol reduces DA transmission by blocking D2 receptors. However, we note that Haloperidol might also increase DA release via action on presynaptic D2 autoreceptors (z.B. *Frank and O'Reilly, 2006*).

We extended previous modeling approaches of exploration behavior (*Daw et al., 2006*; *Speekenbrink and Konstantinidis, 2015*) using a hierarchical Bayesian estimation scheme. Specifically, we jointly examined dopaminergic drug-effects on directed and random exploration as well as perseveration, hypothesizing that choice behavior would be best accounted for by a model that accounts for all three processes (*Schönberg et al., 2007*; *Rutledge et al., 2009*; *Payzan-Lenestour and Bossaerts, 2012*). We hypothesized both random and directed exploration to increase under L-dopa and decrease under haloperidol compared to placebo (*Frank et al., 2009*; *Beeler et al., 2010*; *Gershman and Tzovaras, 2018*). We further hypothesized that this would be accompanied by a corresponding modulation of brain activity in regions implicated in exploration, focusing specifically on FPC, ACC and AI (*Daw et al., 2006*; *Raja Beharelle et al., 2015*; *Blanchard and Gershman, 2018*).

## Results

### Participants learn to keep track of the best bandit

On each testing day, separated by exactly one week, participants performed 300 trials of a four-armed restless bandit task (*Daw et al., 2006*; *Figure 1*; for more details, see Materials and methods section) during fMRI, under three pharmacological conditions (Placebo, Haloperidol, L-DOPA). Overall, participants' choice behavior indicated that they understood the task structure, and tracked the most valuable bandit throughout the task (see *Figure 2*). On trial 1, participants randomly selected one of the four bandits (probability to choose best bandit: 21.5 ± 7.49%, M ± SE). After five trials, participants already selected the most valuable option with 49.03% (±4.98%; M ± SE), which was significantly above chance level of 25% ($t_{30}$ = 4.83, p=3.82*$10^{-5}$, *Figure 2*), and consistently kept choosing the bandit with the highest payoff with on average 67.89% (±2.78%). Thus, participants continuously adjusted their choices to the fluctuating outcomes of the four bandits.

### No significant drug effects on model-free performance measures

We first tested for possible drug effects on model-free measures of task performance. These variables included the overall monetary payout (payout), the percentage of choices of the bandit with the highest actual payoff (%bestbandit, *Figure 2*), the percentage of choice switches (%switches), and median reaction times (median RT). Yet, rmANOVAs yielded no significant drug effect on any of these four model-free choice variables (payout: $F_{2,60}$=0.06, p=0.943; %bestbandit: $F_{2,60}$=0.34, p=0.711; %switches: $F_{2,60}$=1.02, p=0.366; median RT: $F_{2,60}$=0.50, p=0.611).

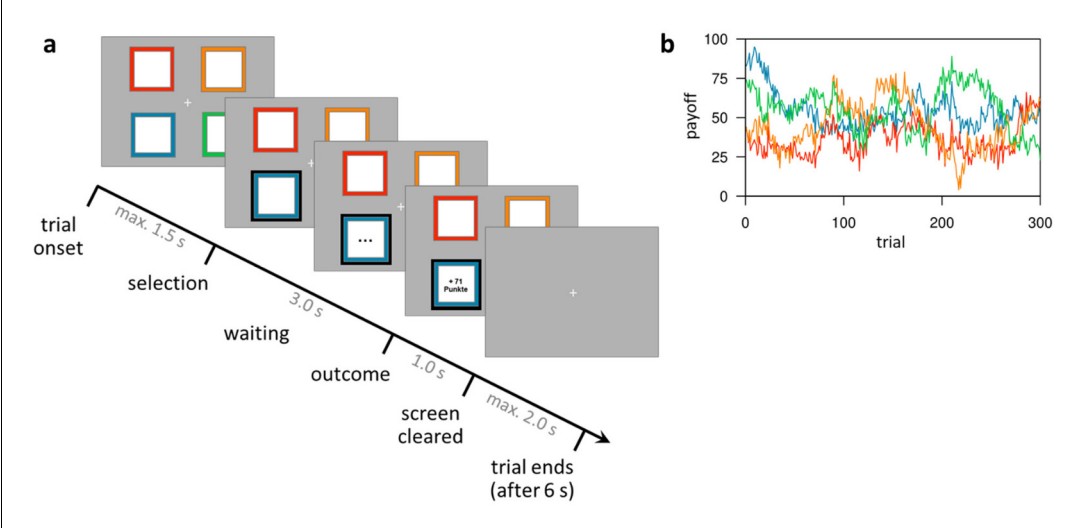

**Figure 1.** Task design of the restless four-armed bandit task (*Daw et al., 2006*). (a) Illustration of the timeline within a trial. At trial onset, four colored squares (bandits) are presented. The participant selects one bandit within 1.5 s, which is then highlighted and, after a waiting period of 3 s, the payoff is revealed for 1 s. After that, the screen is cleared and the next trial starts after a fixed trial length of 6 s plus a variable intertrial interval (not shown) with a mean of 2 s. (b) Example of the underlying reward structure. Each colored line shows the payoffs of one bandit (mean payoff plus Gaussian noise) that would be received by choosing that bandit on each trial.

## Choice behavior contains signatures of directed exploration and perseveration

We then used computational modeling to examine whether choice behavior indeed contained signatures of random exploration, directed exploration and perseveration. To this end, we set up eight separate computational models that differed regarding the implemented learning and choice rules within a hierarchical Bayesian framework using the STAN modeling language (version 2.17.0; *Stan Development Team, 2017*). We compared two learning rules: the classical Delta rule from temporal-difference algorithms (e.g. *Sutton and Barto, 1998*), and a Bayesian learner (*Daw et al., 2006*) that formalizes the updating process with a Kalman filter (*Kalman, 1960*). In the former model, values are updated based on prediction errors that are weighted with a constant learning rate. In contrast, the Kalman filter additionally tracks the uncertainty of each bandit's value, and value updating is proportional to the uncertainty of the chosen bandit (Kalman gain, see Materials and methods section). These learning rules were combined with four different choice rules that were all based on a softmax action selection rule (*Sutton and Barto, 1998*; *Daw et al., 2006*). Choice rule 1 was a standard softmax with a single inverse temperature parameter (β) modeling random exploration. Choice rule 2 included an additional free parameter $\varphi$ modeling an exploration bonus that scaled with the estimated uncertainty of the chosen bandit (directed exploration). Choice rule 3 included an additional free parameter (ρ) modeling a perseveration bonus for the bandit chosen on the previous trial. Finally, choice rule 4 included an additional term to capture random exploration scaling with total uncertainty across all bandits (*Gershman, 2018*). Leave-one-out (LOO) cross-validation estimates (*Vehtari et al., 2017*) were computed over all drug-conditions, and for each condition separately to assess the models' predictive accuracies. The Bayesian learning model with terms for directed exploration and perseveration (Bayes-SMEP) showed highest predictive accuracy in each drug condition and overall (*Figure 3*). The most complex model including an additional total-uncertainty dependent term provided a slightly inferior account of the data compared to the model without this term (loo log-likelihood: Bayes-SME(R)P: -0.5983 (-0.59989)).

## Model-based regressors and classification of exploration trials

In the best-fitting Bayesian model (Bayes-SMEP), participants' choices are stochastically dependent on three factors: the prior belief of the mean reward value of each bandit ($\mu^{pre}$; *Figure 4a*), the exploration bonus, that is the prior belief of each bandit's payout variance ('uncertainty') scaled with

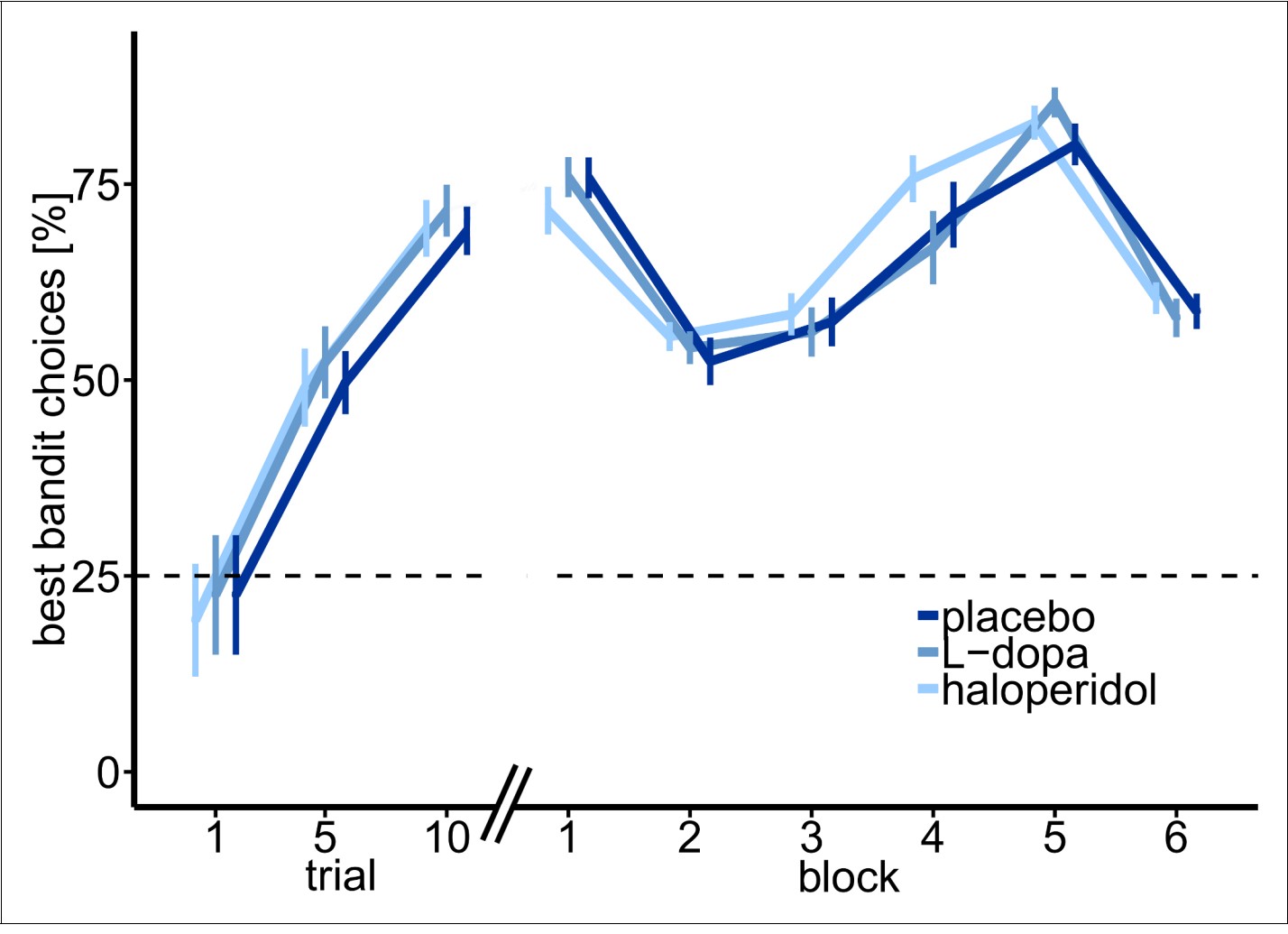

**Figure 2.** Percentage of optimal choices (highest payoff) throughout the task. Shown are the mean percentage of choosing the best bandit in trials 1–10, and over task blocks of trials 11–50 (block 1) and 51–300 separated in 5 blocks of 50 trials each, over all participants, and for each drug session separately. Participants started with randomly (~25%) choosing one bandit in trial 1 (21.5% ± 7.49%, M ± SE). After five trials participants already chose the most valuable bandit with 49.03 ± 4.98% (M ± SE).

the exploration bonus parameter $\varphi$ ($\varphi\sigma^{pre}$; *Figure 4b*), and the perseveration bonus (I$\rho$; where $I$ denotes an indicator function with respect to the bandit chosen on the previous trial; *Figure 4c*). Based on these quantities, which are computed for each bandit, the model computes the choice probabilities for all four bandits on each trial ($P$; *Figure 4d*, See *Equation 7* in the Materials and methods section). Between trials, participants' prior belief of the chosen bandit's mean reward value is updated according to the reward prediction error ($\delta$, *Figure 4e*), as the difference between their prior belief and the actual reward outcome of the chosen bandit. Based on the model, participants' choices can be classified as exploitation (i.e. when the bandit with highest expected value was selected), or exploration (i.e. when any other bandit was selected) (*Daw et al., 2006*). We extended this binary classification of *Daw et al., 2006* by further dividing exploration trials into directed exploration (i.e. trials where the bandit with the highest exploration bonus was chosen), and random exploration trials (i.e. trials where one of the remaining bandits was chosen). The trinary classification scheme corresponded well with the respective model parameters ($\beta$-random exploration, $\varphi$-directed exploration, $\rho$-perseveration; *Appendix 1—figure 1*). Over trials, the summed uncertainty over all bandits ($\Sigma\sigma^{pre}$; *Figure 4f*) fluctuates in relation to the fraction of exploration.

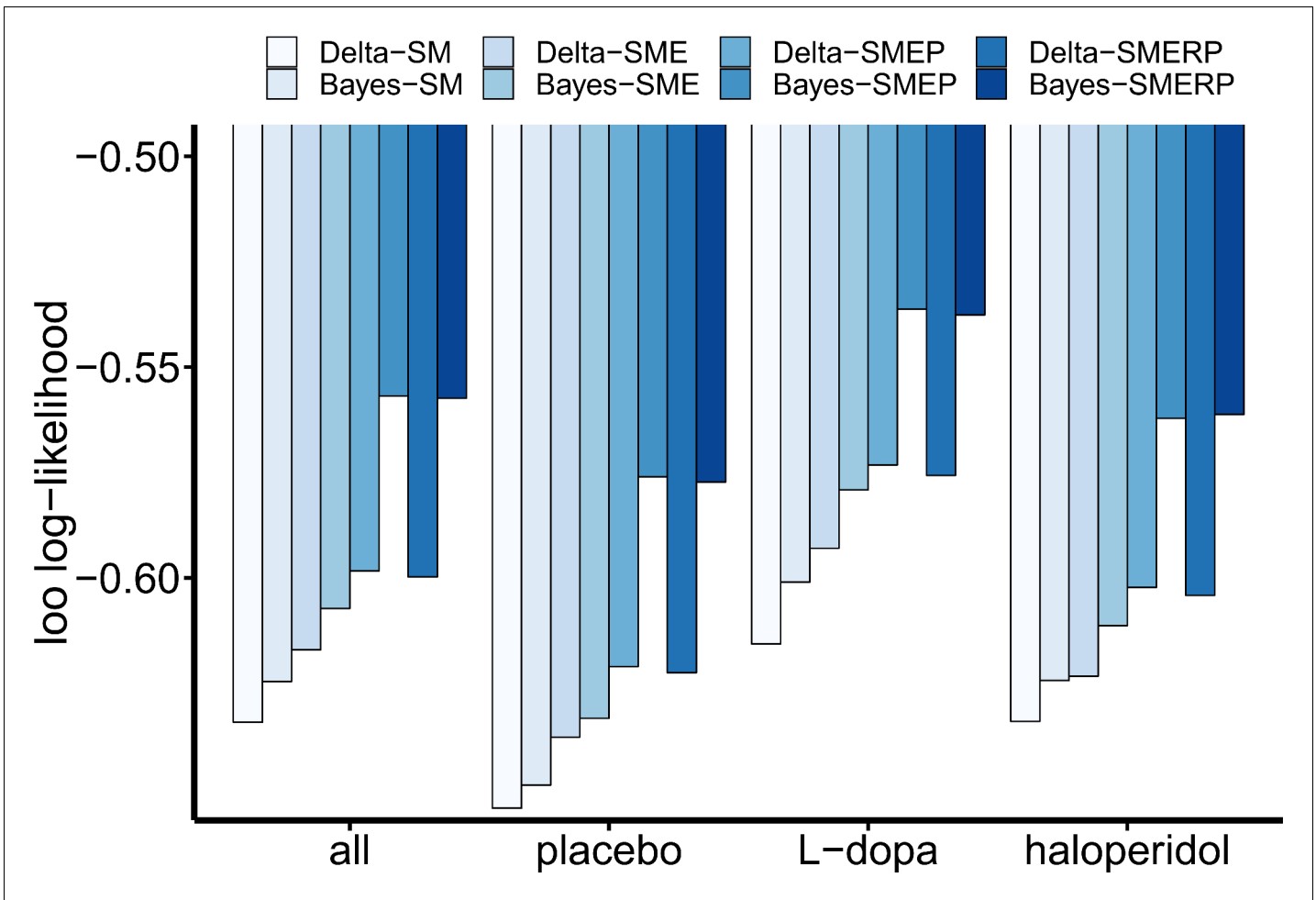

**Figure 3.** Results of the cognitive model comparison. Leave-one-out (LOO) log-likelihood estimates were calculated over all drug conditions (n = 31 subjects with t = 3*300 trials) and once separately for each drug condition (n = 31 with t = 300). All LOO estimates were divided by the total number of data points in the sample (n*t) for better comparability across the different approaches. Note that the relative order of LOO estimates is invariant to linear transformations. Delta: simple delta learning rule; Bayes: Bayesian learner; SM: softmax (random exploration); E: directed exploration; R: total uncertainty-based random exploration; P: perseveration.

## L-dopa reduces directed exploration

Next, we tested for possible drug effects on the percentage of exploitation and exploration trials (overall, random and directed) per subject. Three separate rmANOVAs with within factors drug and trial (6 blocks of 50 trials each) were computed for each of the following four dependent variables: the percentage of (a) exploitation trials, (b) random exploration trials, and (c) directed exploration trials. We found a significant drug effect only for the percentage of directed explorations ($F_{1.66,49.91}=7.18$, p=.003; *Figure 5c*). The percentage of random explorations ($F_{2,60}=0.55$, p=.58, *Figure 5b*) or exploitations ($F_{2,60}=1.57$, p=.22; *Figure 5a*) were not significantly modulated by drug. All drug $\times$ trial interactions were not significant (p>=0.19). Post-hoc, paired t-tests showed a significant reduction in the percentage of directed explorations under L-dopa compared to placebo (mean difference P-D=2.82, $t_{30}=4.69$, p<.001) and haloperidol (mean difference H-D = 2.42, $t_{30}=2.76$, p=.010), but not between placebo and haloperidol (mean difference P-H=0.39, $t_{30}=0.43$, p=.667). Notably, an exploratory t-test revealed that the percentage of exploitations was marginally increased under L-dopa compared to placebo (mean difference P-D=-2.61, $t_{30}=-1.92$, p=.065).

Importantly, the observed attenuation of directed exploration trials under L-dopa was mirrored in our analysis of drug effects on the model parameters' posterior distributions. Dopaminergic drug effects were first examined for the group-level (mean $M$ posteriors for β (random exploration), $\varphi$

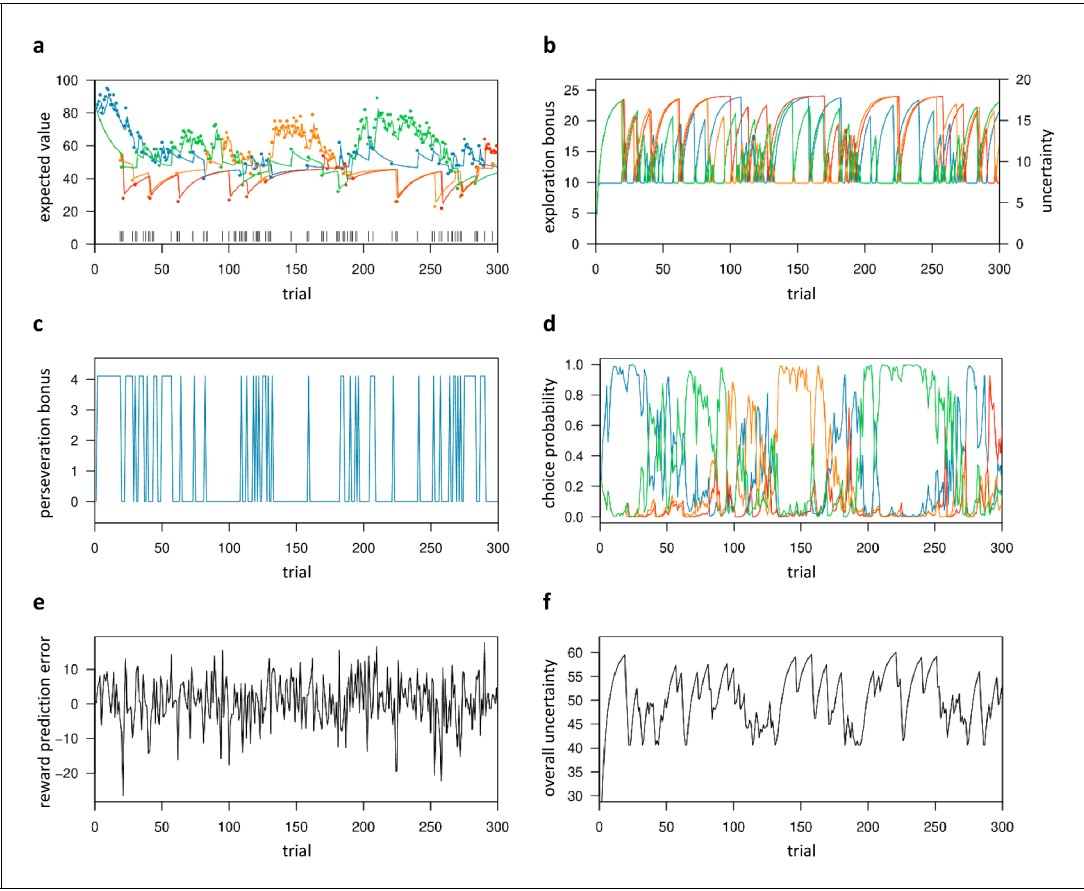

**Figure 4.** Trial-by-trial variables of the best-fitting Bayesian model (Bayes-SMEP). Trial-by-trial estimates are shown for the placebo data of one representative subject with posterior medians: β=0.29, $\varphi$ = 1.34, and ρ=4.11 (random exploration, directed exploration, and perseveration). (a) Colored lines depict the expected values ($\mu^{pre}$) of the four bandits, whereas colored dots denote actual payoffs. Vertical black lines mark trials classified as exploratory (*Daw et al., 2006*). (b) Exploration bonus ($\varphi\sigma^{pre}$) and uncertainty ($\sigma^{pre}$) for each bandit. (c) Perseveration bonus (Iρ). This bonus is a fixed value added only to the bandit chosen in the previous trial, shown here for one bandit. (d) Choice probability (*P*). Each colored line represents one bandit. (e) Reward prediction error (δ). (f) The subject's overall uncertainty ($\Sigma\sigma^{pre}$), that is the summed uncertainty over all four bandits.

(directed exploration) and ρ (perseveration)) of the best-fitting Bayesian model, which were estimated separately for each drug. In accordance with the drug effects on fraction of directed explorations, under L-dopa, the posterior group-level mean of $\varphi$ ($M^\varphi$) was substantially reduced compared to both placebo and haloperidol (*Figure 6*), such that the 90% highest density intervals of the difference of the posterior distributions of $M^\varphi$ did not overlap with zero (*Appendix 1—figure 1b*). In contrast, we did not observe effects of L-dopa on random exploration (β, *Figure 6*) or perseveration (ρ, *Figure 6*). Somewhat surprisingly, haloperidol showed no effects on the posterior group-level means of any parameter.

In summary, we found that boosting central dopamine with L-dopa specifically attenuated exploratory choice patterns that aimed at reducing uncertainty of highly uncertain choice options. At the same time, exploitative choices were marginally increased. Although we observed no significant drug × trial interactions, the L-dopa-induced effects appeared to be more pronounced in the first third of the task (*Figure 5a, c*).

## Distinct brain networks orchestrate exploration and exploitation

Analysis of the imaging data proceeded in two steps. First, we examined our data for overall effects of exploration/exploitation based on a binary trial classification, as well as on the model-based parametric effects of expected value and uncertainty. In a second step, we examined the neural basis of the drug-induced change in exploration. All reported fMRI results are based on statistical parametric maps (SPMs) thresholded at p<0.05, FWE-corrected for whole brain volume (unless

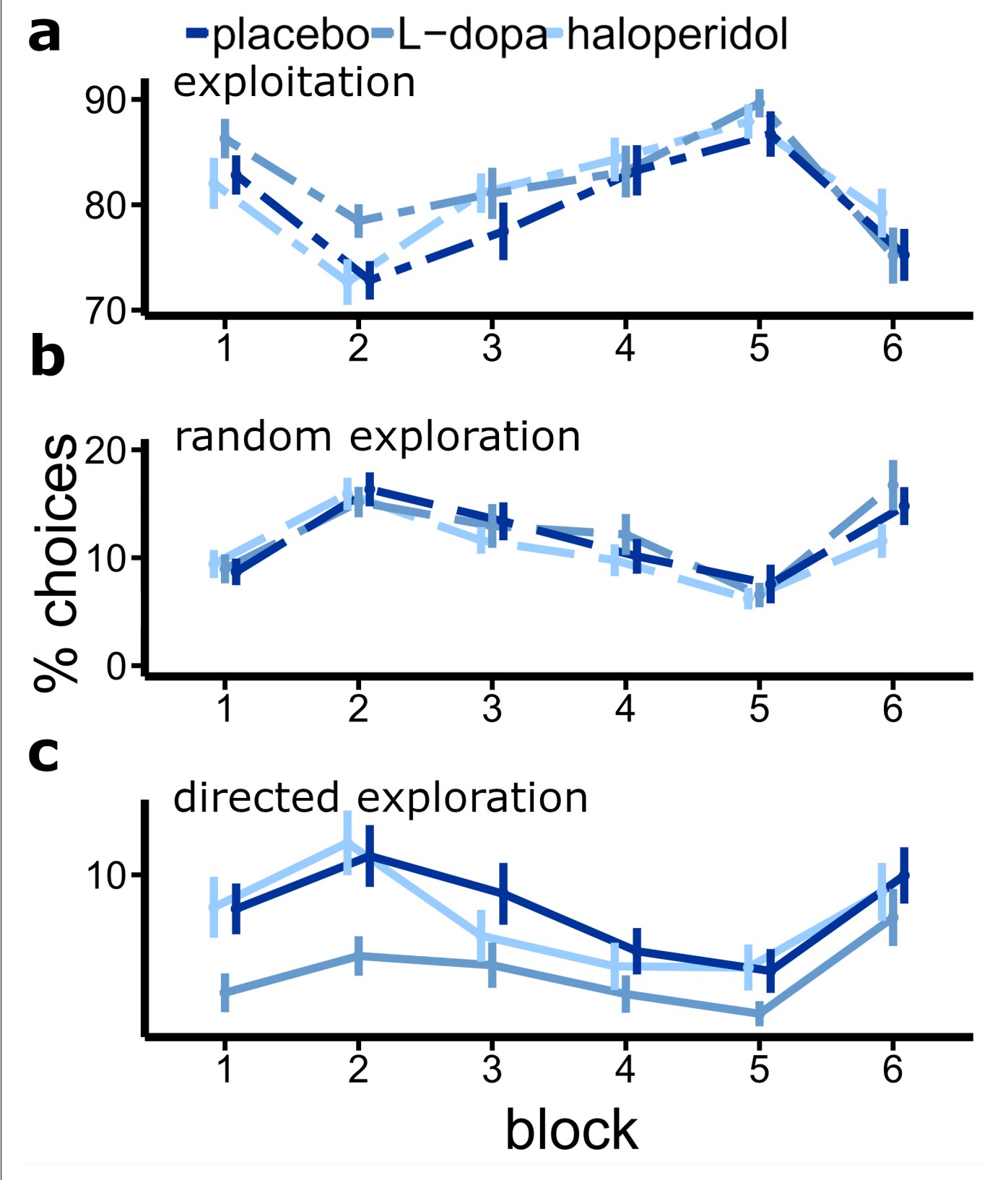

**Figure 5.** Drug effects on the percentage of exploitations and explorations (bandit with highest uncertainty is chosen). Shown are the mean percentage of directed explorations for each drug session over six blocks of 50 trials each (error bars indicate standard error of the mean).

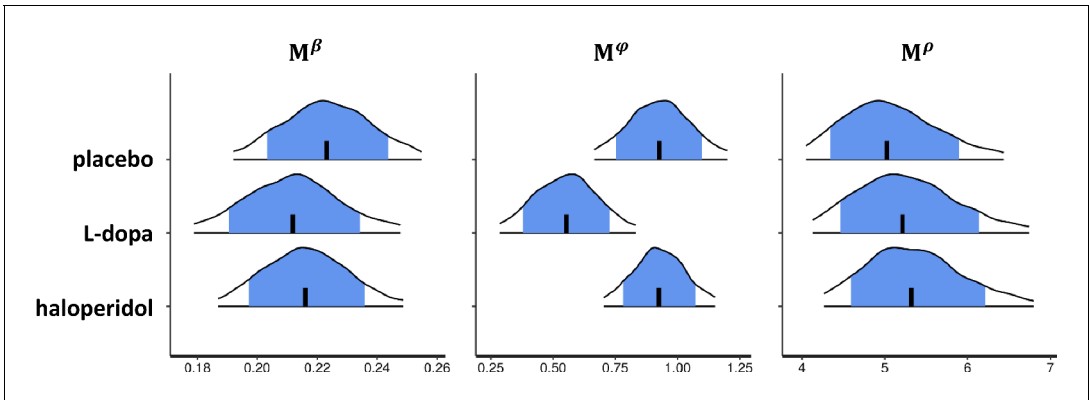

**Figure 6.** Drug effects for the group-level parameter estimates of the best-fitting Bayesian model. Shown are posterior distributions of the group-level mean ($M$) of all choice parameters ($\beta$, $\varphi$, $\rho$), separately for each drug condition. Each plot shows the median (vertical black line), the 80% central interval (blue (grey) area), and the 95% central interval (black contours); $\beta$: random exploration, $\varphi$: directed exploration; $\rho$: perseveration parameter. For drug effects on the standard deviation of the group-level median parameters $\varphi$, $\beta$ and $\rho$ see *Appendix 1—figure 1a*. See *Appendix 1—figure 1b and c* for pairwise drug-related differences of the group-level mean (M) and (c) standard deviation ($\Lambda$) of $\varphi$.

stated otherwise, for example for visualization purposes). In addition to whole-brain analyses, drug-induced changes were additionally assessed in a regions of interest approach based on small volume FWE correction (p<0.05) for seven regions that have previously been associated with exploration: the left/right FPC and left/right IPS (*Daw et al., 2006*), as well as the dACC and left/right AI (*Blanchard and Gershman, 2018*). Regions used for small volume correction were defined by 10 mm radius spheres centered at peak voxels reported in these previous studies (*Appendix 1—table 5*).

In a first general linear model (GLM), differences in brain activity between exploratory and exploitative choices were modeled (at trial onset) across all participants and drug conditions, using the binary classification previously described by *Daw et al., 2006*. In accordance with previous work, the pattern of brain activity differed markedly between both types of choices (*Figure 7*, *Appendix 1—figure 5*). Highly similar activation patterns were found with a second GLM that was based on the parametric regressors expected value ($\mu^{pre}$) and uncertainty ($\sigma^{pre}$) of the chosen bandit, both modeled at trial onset. While expected value related neural activity largely overlapped with the one

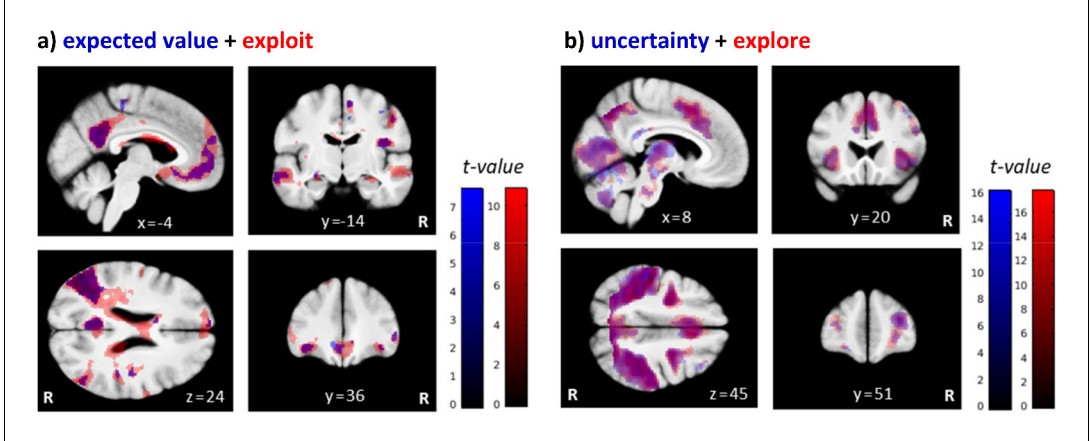

**Figure 7.** Brain regions differentially activated by exploratory and exploitative choices. Shown are overlays of statistical parametric maps (SPMs) for the contrast (a) the parametric regressor expected value ($\mu^{pre}$) of the chosen bandit (in blue) and the binary trial classification related contrast exploit > explore ('exploit' in red), and for (b) the parametric regressor uncertainty ($\sigma^{pre}$) (in blue) and the contrast explore > exploit ('explore' in red), over all drug conditions. For visualization purposes: thresholded at p<0.001, uncorrected. R: right.

**Table 1.** Free and fixed parameters of all six computational models.

| | Delta rule | | Bayes learner rule | |
|---|---|---|---|---|
| Choice rule 1 | $\alpha, \beta$ | fixed: $v_1$ | $\beta$ | fixed: $\hat{\lambda}, \hat{\vartheta}, \hat{\sigma}_0^2, \hat{\sigma}_d^2, \hat{\mu}_1^{pre}, \hat{\sigma}_1^{pre}$ |
| Choice rule 2 | $\alpha, \beta, \varphi$ | fixed: $v_1$ | $\beta, \varphi$ | fixed: $\hat{\lambda}, \hat{\vartheta}, \hat{\sigma}_0^2, \hat{\sigma}_d^2, \hat{\mu}_1^{pre}, \hat{\sigma}_1^{pre}$ |
| Choice rule 3 | $\alpha, \beta, \varphi, \rho$ | fixed: $v_1$ | $\beta, \varphi, \rho$ | fixed: $\hat{\lambda}, \hat{\vartheta}, \hat{\sigma}_0^2, \hat{\sigma}_d^2, \hat{\mu}_1^{pre}, \hat{\sigma}_1^{pre}$ |
| Choice rule 4 | $\alpha, \beta, \varphi, \rho, \gamma$ | fixed: $v_1$ | $\beta, \varphi, \rho, \gamma$ | fixed: $\hat{\lambda}, \hat{\vartheta}, \hat{\sigma}_0^2, \hat{\sigma}_d^2, \hat{\mu}_1^{pre}, \hat{\sigma}_1^{pre}$ |

*Note*: Free parameters are only listed for the subject-level. Hierarchical models contained for each free subject-level parameter $x$ two additional free parameters ($M^x, \Lambda^x$) on the group-level (**Figure 9**). Choice rule 1: softmax; Choice rule 2: softmax with exploration bonus; Choice rule 3: softmax with exploration bonus and perseveration bonus; $\alpha$: learning rate; $\beta$: softmax parameter; $\varphi$: exploration bonus parameter; $\rho$: perseveration bonus parameter; , $\gamma$: uncertainty-based random exploration parameter; $v_1$: initial expected reward values for all bandits; $\lambda$: decay parameter; $\vartheta$: decay center; $\sigma_o^2$: observation variance; $\sigma_d^2$: diffusion variance; $\mu_1^{pre}$: initial mean of prior expected rewards for all bandits; $\sigma_1^{pre}$: initial standard deviation of prior expected rewards for all bandits.

for exploitative choices (**Figure 7a**), uncertainty associated patterns of neural activation overlapped with the ones for exploratory choices (**Figure 7b**).

Replicating earlier findings (e.g. **Daw et al., 2006**; **Addicott et al., 2014**), exploration trials were associated with greater activation in bilateral frontopolar cortex (FPC; left: −42, 27, 27 mm; z = 6.07; right: 39, 34, 28 mm; z = 7.56), in a large cluster along the bilateral intraparietal sulcus (IPS; cluster peak at −48,−33, 52; z = 10.45), and in bilateral anterior insula (AI; left: −36, 15, 3 mm; z = 6.69; right: 36, 20, 3 mm; z = 6.87) as well as dorsal anterior cingulate cortex (dACC; cluster peak at 8, 12, 45 mm; z = 8.47). Clusters within thalamus, cerebellum, and supplementary motor area also showed increased bilateral activation during exploration compared to exploitation.

In contrast, exploitative choices were associated with greater activation in the ventromedial prefrontal cortex (vmPFC; −2, 40,−10 mm; z = 5.67) and in bilateral lateral orbitofrontal cortex (lOFC; left: −38, 34,−14 mm; z = 5.81; right: 38, 36,−12 mm; z = 5.02). Furthermore, greater activation during exploitative trials was also observed in a cluster spanning the left posterior cingulate cortex (PCC) and left precuneus (cluster peak at −6,−52, 15 mm; z = 7.40), as well as in the angular gyrus (left: −42,−74, 34 mm; z = 8.04; right: 52,−68, 28 mm; z = 7.02), hippocampus (left: −24,−16, −15 mm; z = 4.16; only at p<0.001, uncorrected; right: 32,−16, −15 mm; z = 5.09), and several clusters along the superior and middle temporal gyrus. A complete list of activations associated with explorative and exploitative choices, as well as activation related to model-based PEs can be found in Appendix 1. We observed no differential activation patterns for directed and random exploration types when expanding the original trial classification of **Daw et al., 2006** in a third GLM (**Appendix 1—figure 6**).

## No evidence for a direct drug modulation of exploration/exploitation-related brain activation

To test for a main effect of drug on differential explore/exploit-related brain activation, we conducted rmANOVAs on the second level contrasts explore vs. exploit of the first GLM as well as on the contrasts related to the parametric regressors of expected value and uncertainty of the second GLM. Surprisingly, we found no suprathreshold activations on the whole-brain level, nor in any of seven regions of interest (ROIs) with small volume correction applied (i.e. left/right FPC, left/right IPS, left/right AI, and dACC).

The same analyses were run for the contrasts from the third GLM, that is directed exploration vs. exploit, random exploration vs. exploit, and random vs. directed exploration. Again, we found no significant drug-related modulation of brain activation for these contrasts in any of our a priori ROIs. Likewise, further analyses revealed no drug effects on neural activation (trial onset, reward onset, PE, and outcome value; see Appendix 1 for details) and no evidence for an association of drug effects with putative proxy measures of baseline DA availability (spontaneous blink rate and working memory span, see Appendix 1 for details).

# L-dopa indirectly modulates exploration via reducing neural coding of overall uncertainty

Based on the null findings in the planned analyses, we reasoned that L-dopa might attenuate directed exploration not by modulating brain activation for exploratory/exploitative choices in a direct manner, but rather by modulating neural computations that are involved in switching from exploitation to directed exploration. Thus, L-dopa might delay the time point at which directed exploration is triggered in response to accumulating 'total' (summed) uncertainty. As we did not find evidence (according to model comparison, see *Figure 3*) for total (summed) uncertainty ($\Sigma\sigma^{pre}$) related random exploration, this was not accounted for in our final Bayesian model. Still, total uncertainty is linked (via the exploration bonus parameter $\varphi$) to the probability to explore a previously unchosen bandit. $\Sigma\sigma^{pre}$ gradually increases during a series of exploitations but reduces abruptly when a bandit with high uncertainty is explored (see *Figure 2f*). We therefore included model-based overall uncertainty as a parametric regressor modeled at trial onset in a new GLM to reveal brain activation that tracks accumulating uncertainty during learning. The contrast images for this regressor were then used in a second-level random effects analysis with the factors drug condition and subject. In the placebo condition alone, no voxels survived whole-brain FWE correction (p<.05), but a more lenient threshold (p<.001, uncorrected) revealed activity in the bilateral dACC (cluster peak at -3, 21, 39mm; z=3.96), right anterior insula (42, 15, -6mm; z=3.46), and left posterior insula (PI) (-34, -20, 8mm; z=4.63) that was positively correlated with the overall uncertainty (*Figure 8a*, *Table 2*). To test our exploratory hypothesis, we computed directed t-contrasts for L-dopa vs. placebo (placebo>L-dopa and L-dopa>placebo). While the contrast L-dopa>placebo yielded no suprathreshold activations, the reverse contrast (placebo>L-dopa) revealed a significant activation in the left PI (-34, -20, 8mm; z=5.05). At a reduced threshold (p<.001, uncorrected), left AI (-38, 6, 14mm; z=4.88) and bilateral dACC (left: -2, 36, 33mm; z=3.32; right: 4, 14, 28mm; z=3.41) showed a stronger correlation with the overall uncertainty under placebo compared to L-dopa (*Figure 8b*, *Table 2*).

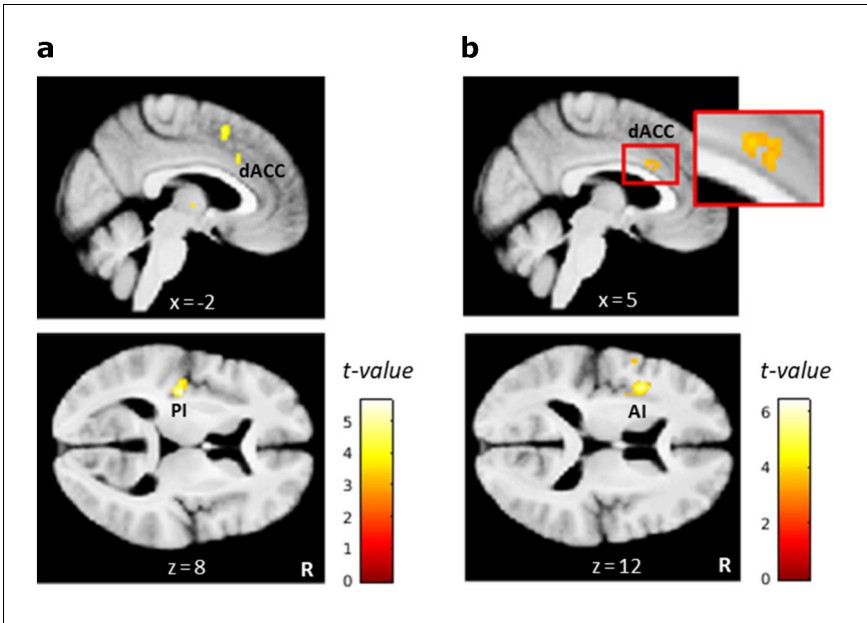

**Figure 8.** L-dopa effects on neural coding of overall uncertainty. (a) Regions in which activity correlated positively with the overall uncertainty in the placebo condition included the dorsal anterior cingulate cortex (dACC) and left posterior insula (PI). (b) Regions in which the correlation with overall uncertainty was reduced under L-dopa compared to placebo included the dACC and left anterior insula (AI). Thresholded at p<0.001, uncorrected. R: right.

**Table 2.** Brain regions in which activity was significantly correlated with the overall uncertainty (fourth GLM), shown for the placebo condition and for pairwise comparison with L-dopa.

| Region | MNI coordinates | | | peak | cluster |
|---|---|---|---|---|---|
| | x | y | z | z-value | extent (k) |
| Placebo | | | | | |
| L posterior insula | −34 | −20 | 8 | 4.63 | 198 |
| R supplementary motor cortex | 8 | 10 | 52 | 3.98 | 92 |
| R/L dorsal anterior cingulate cortex, L supplementary motor cortex | -3 | 21 | 39 | 3.96 | 176 |
| R anterior insula | 42 | 15 | -6 | 3.46 | 38 |
| R thalamus | 8 | −10 | 2 | 3.41 | 18 |
| Placebo > L-dopa | | | | | |
| L posterior insula | −34 | −20 | 8 | 5.05* | 82 |
| L anterior insula, L frontal operculum | −38 | 6 | 14 | 4.88 | 222 |
| L opercular part of the inferior frontal gyrus | −42 | 9 | 26 | 4.01 | 80 |
| L precentral gyrus | −54 | 3 | 12 | 3.47 | 23 |
| R dorsal anterior cingulate cortex | 4 | 14 | 28 | 3.41 | 32 |
| R precentral gyrus | 39 | -9 | 44 | 3.39 | 16 |
| L dorsal anterior cingulate cortex | -2 | 36 | 33 | 3.32 | 17 |
| L-dopa > placebo | | | | | |
| no suprathreshold activation | | | | | |

Note: Thresholded at p<0.001, uncorrected, with k ≥ 10 voxels; L: left; R: right.
*p=0.031, FWE-corrected for whole-brain volume.

## Discussion

Here, we directly tested the causal role of DA in human explore/exploit behavior in a pharmacological, computational fMRI approach, using L-dopa (DA precursor) and haloperidol (DA antagonist) in a double-blind, placebo-controlled, counterbalanced, within-subjects design. Model comparison revealed that choice behavior was best accounted for by a novel extension of a Bayesian learning model (*Daw et al., 2006*), that included separate terms for directed exploration and choice perseveration. Modeling revealed that directed exploration was reduced under L-dopa compared to placebo and haloperidol. In contrast, no drug effects were observed on parameters capturing random exploration (β) or perseveration (ρ). On the neural level, exploration was associated with higher activity in the FPC, IPS, dACC, and insula, whereas exploitation showed higher activity in the vmPFC, OFC, PCC, precuneus, angular gyrus, and hippocampus, replicating previous studies (*Daw et al., 2006*; *Addicott et al., 2014*; *Blanchard and Gershman, 2018*). Surprisingly, no drug effects were found for these effects, nor on striatal reward prediction error signaling (see Appendix 1 for details). However, an exploratory model-based analysis revealed that L-dopa reduced insular and dACC activity associated with total (summed) uncertainty.

### Computational modeling of exploration

We examined two learning rules (Delta rule vs. Bayesian learner) and four choice rules resulting in a total of eight computational models. Model comparison revealed that the Bayesian learning model (Kalman Filter) outperformed the Delta rule for each of the choice rules. Although both learning rules are based on the same error-driven learning principle, the Bayesian learner assumes that subjects additionally track the variance (uncertainty) of reward expectation and adjust the learning rate from trial to trial according to the current level of uncertainty - learning is high when reward predictions are uncertain (i.e. during exploration), but decreases when predictions become more accurate (i.e. during exploitation).

For both learning rules the model including separate parameters for random exploration, directed exploration and perseveration accounted for the data best, in line with recent work from

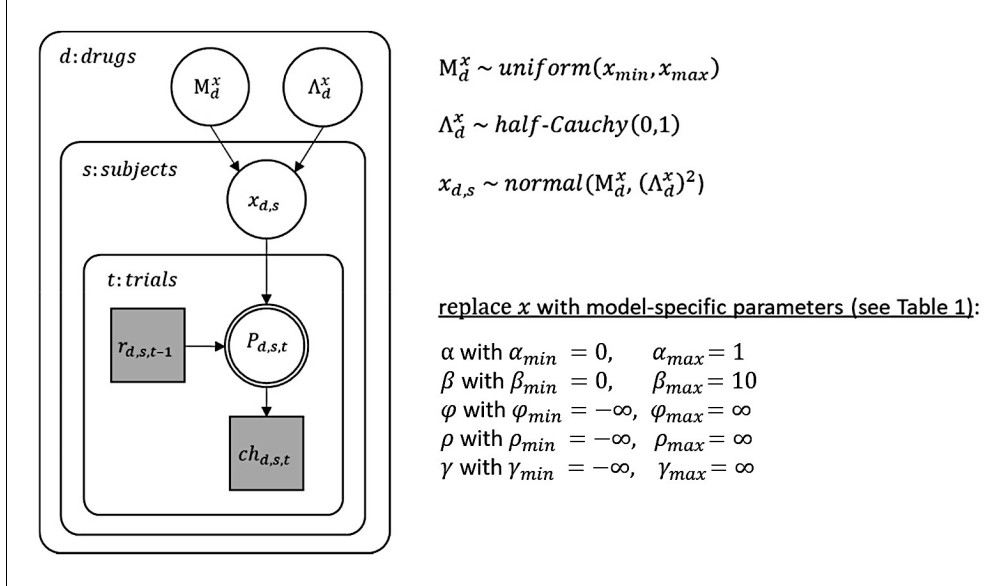

**Figure 9.** Graphical description of the hierarchical Bayesian modeling scheme. In this graphical scheme, nodes represent variables of interest (squares: discrete variables; circles: continuous variables) and arrows indicate dependencies between these variables. Shaded nodes represent observed variables, here rewards ($r$) and choices ($ch$) for each trial ($t$), subject ($s$), and drug condition ($d$). For each subject and drug condition, the observed rewards until trial t-1 determine (deterministically) choice probabilities ($P$) on trial $t$, which in turn determine (stochastically) the choice on that trial. The exact dependencies between previous rewards and choice probabilities are specified by the different cognitive models and their model parameters ($x$). Note that the double-bordered node indicates that the choice probability is fully determined by its parent nodes, that is the reward history and the model parameters. As the model parameters differ between all applied cognitive models, they are indicated here by an $x$ as a placeholder for one or more model parameter(s). Still, the general modeling scheme was the same for all models: Model parameters were estimated for each subject and drug condition and were assumed to be drawn from a group-level normal distribution with mean $M^x$ and standard deviation $\Lambda^x$ for any parameter $x$. Note that group-level parameters were estimated separately for each drug condition. Each group-level mean ($M^x$) was assigned a non-informative (uniform) prior between the limits $x_{min}$ and $x_{max}$ as listed above. Each group-level standard deviation ($\Lambda^x$) was assigned a half Cauchy distributed prior with location parameter 0 and scale 1. Subject-level parameters included $\alpha, \beta, \varphi, \rho,$ and $\gamma$ depending on the cognitive model (see *Table 1*).

our group (*Wiehler et al., 2019*). Note that we also replicated this effect in the original data from the *Daw et al., 2006* paper (*Wiehler et al., 2019*). Random exploration was implemented via adding stochasticity to the action selection with a softmax formulation (*Thrun, 1992*; *Daw et al., 2006*; *Beeler et al., 2010*; *Gershman, 2018*). Directed exploration was implemented via an exploration bonus parameter (e.g. *Dayan and Sejnowski, 1996*; *Daw et al., 2006*). In this scheme, 'uncertainty' or 'information' biases choices toward more uncertain/informative options by increasing their value. In contrast to *Gershman and Tzovaras, 2018*, we found no evidence in our data that including an additional term for total-uncertainty based exploration further improved model fit.

## Behavioral DA drug effects

Overall, our finding that pharmacological manipulation of the DA system impacts the exploration/exploitation trade-off is in line with previous animal and human studies (e.g. *Frank et al., 2009*; *Beeler et al., 2010*; *Blanco et al., 2015*). However, the observed pattern of drug effects did not match our initial hypothesis, according to which both random and directed exploration were expected to *increase* under L-dopa vs. placebo and *decrease* under haloperidol vs. placebo.

### L-dopa

L-dopa administration reduced directed exploration ($\varphi$) compared to placebo, while random exploration (β) was unaffected. In the model, this could reflect (1) a reduced tendency for directed

exploration and/or (2) an increased tendency for value-driven exploitation. Accordingly, when classifying all choices per subject into exploitations, directed exploration and random exploration, L-dopa was found to reduce the percentage of directed but not random exploration compared to placebo across subjects, and marginally increase the percentage of exploitations.

Previous studies that found that DA promotes exploration primarily focused on prefrontal DA availability (*Frank et al., 2009*; *Blanco et al., 2015*; *Kayser et al., 2015*) or examined effects of tonic DA modulation (*Beeler et al., 2010*; *Costa et al., 2014*). L-Dopa, however, likely increases DA transmission most prominently within striatum, and to a much lesser degree in PFC (*Carey et al., 1995*; *Cools, 2006*). PET studies in humans also indicate that L-dopa primarily increases phasic rather than tonic striatal DA activity (*Floel et al., 2008*; *Black et al., 2015*). Increased phasic DA release was associated with improved learning under L-dopa compared to placebo, presumably by enhancing the reinforcing effect of positive feedback during learning (*Frank et al., 2004*; *Pessiglione et al., 2006*; *Cox et al., 2015*; *Mathar et al., 2017*). Thus, L-dopa might strengthen-positive reinforcing effects of immediate rewards (*Pine et al., 2010*) via increased phasic striatal DA release, fostering both impulsive and exploitative choice behavior (*Kobayashi and Schultz, 2008*; *Schultz, 2010*; *Pine et al., 2010*).

In addition, the striatum is densely interconnected with frontal cortices (*Haber and Knutson, 2010*; *Badre and Nee, 2018*). Thus, L-dopa might have also attenuated exploration via indirectly modulating frontal activity through bottom-up interference (*Kellendonk et al., 2006*; *Simpson et al., 2010*; *Kohno et al., 2015*; *Duvarci et al., 2018*).

### Haloperidol

Contrary to our hypothesis, we did not observe a significant modulation of exploration/exploitation under haloperidol. Computational modeling revealed no changes in the group-level mean parameters for directed exploration ($\varphi$), random exploration ($\beta$) or perseveration.

Haloperidol is a potent D2 receptor antagonist. Thus, the absence of a clear effect on exploration and/or exploitation is at first glance somewhat puzzling. We predicted haloperidol to attenuate tonic DA signaling and reduce directed and random exploration. However, alternatively one could have predicted haloperidol to reduce phasic signaling within striatum and attenuate reward exploitation and thus increase exploration (*Pessiglione et al., 2006*; *Pleger et al., 2009*; *Eisenegger et al., 2014*).

In addition, numerous studies found opposite effects of single doses of haloperidol, and the directionality of effects might depend on the dosage (*Frank and O'Reilly, 2006*; *Jocham et al., 2011*; *Frank and O'Reilly, 2006*; *Pine et al., 2010*). Low doses of D2 antagonists can stimulate DA signaling, possibly via acting on presynaptic auto-receptors (*Schmitz et al., 2003*; *Frank and O'Reilly, 2006*; *Ford, 2014*), which contrasts with the antidopaminergic effects observed under chronic and high-dose treatment (*Starke et al., 1989*; *Frank and O'Reilly, 2006*; *Knutson and Gibbs, 2007*).

Finally, a potential baseline dependency may have given rise to a complex non-linear pattern of drug effects. However, we also examined two proxy measures for striatal DA transmission (spontaneous eye blink rate and working memory capacity), both of which showed no modulation of drug-effects (see Appendix 1).

## fMRI findings
### Neural correlates of exploration and exploitation

Consistent with previous research in primates and humans (*Ebitz et al., 2018*; *Daw et al., 2006*), the pattern of brain activity differed markedly between exploration and exploitation, both using a binary classification of trials and via parametric model-based analyses: Exploratory choices were associated with higher activity in the FPC, IPS, dACC, and AI, replicating previous human fMRI studies (*Daw et al., 2006*; *Addicott et al., 2014*; *Laureiro-Martínez et al., 2014*; *Laureiro-Martínez et al., 2015*; *Zajkowski et al., 2017*). It has been suggested that the FPC may track information relevant for exploratory decisions, such as the expected reward and uncertainty of unchosen choice options, and trigger a behavioral switches from an exploitative to an exploratory mode whenever the accumulated evidence supports such a switch (*Boorman et al., 2009*; *Boorman et al., 2011*; *Badre et al., 2012*; *Cavanagh et al., 2012*). The IPS, in contrast, has been suggested to serve

as an interface between frontal areas and motor output areas, initiating behavioral responses to implement exploratory actions (*Daw et al., 2006*; *Boorman et al., 2009*; *Laureiro-Martínez et al., 2015*). The dACC and AI, on the other hand, are thought to form a salience network involved in detecting and orienting toward salient stimuli (*Menon, 2015*; *Uddin, 2015*), which may also sub-serve attentional and behavioral switching from exploitation to exploration.

Exploitative choices were associated with greater activation in vmPFC and OFC, again replicating previous work (*Laureiro-Martínez et al., 2014*; *Laureiro-Martínez et al., 2015*). Both regions are implicated in coding the subjective value of attainable goods (*Levy and Glimcher, 2012*). Thus, vmPFC and OFC might foster exploitation based on computing subjective values of decision options (*Kringelbach and Rolls, 2004*; *O'Doherty, 2004*; *O'Doherty, 2011*; *Peters and Büchel, 2010*; *Bartra et al., 2013*).

In addition, greater activation during exploitation was also observed in the PCC, angular gyrus, precuneus, and hippocampus, partly replicating the results of earlier studies (*Addicott et al., 2014*; *Laureiro-Martínez et al., 2014*; *Laureiro-Martínez et al., 2015*). Together with the medial PFC, these regions are hypothesized to form a large-scale brain system referred to as the 'default mode network' (DMN; *Raichle et al., 2001*; *Andrews-Hanna et al., 2014*). Thus, activity within these regions during exploitation may also relate to a reduced cognitive and attentional demand during exploitation compared to exploration. The angular gyrus is also heavily implicated in number moni-toring (*Göbel et al., 2001*) and thus may monitor reward values during exploitation (see *Addicott et al., 2014*). The PCC is considered to be part of the brain's valuation system and may encode reward-related information during exploitation (*Lebreton et al., 2009*; *Bartra et al., 2013*; *Grueschow et al., 2015*), although in primates PCC neurons were shown to signal exploratory deci-sions (*Pearson et al., 2009*). A further characterization of the hypothesized functions of specific sub-regions of the exploitation- and exploration networks naturally requires direct experimental tests in the future.

## Neural DA drug effects

We found no direct drug effects on exploration- or exploitation-related brain activity, nor on the neural correlates of reward PE signals (see Appendix 1). We hypothesized DA drug effects on exploratory behavior to be associated with changes in the activity of brain regions implicated in exploratory choices. This was not supported by the fMRI data. Alternatively (or additionally), the observed L-dopa effect on explore/exploit behavior could also be due to an enhanced phasic DA release and PE signaling in the striatum (see above). In such a model, L-dopa would be expected to increase the magnitude of the striatal reward PE signal, as previously shown by *Pessiglione et al., 2006*. However, a recent study (*Kroemer et al., 2019*) did not observe such a modulatory effect of L-dopa on PE coding in a sequential reinforcement learning task in a large community sample (N = 65).

Several factors might have contributed to the absence of significant L-dopa effects on the neural correlates of explore/exploit decisions or the reward PE. First, this failure may simply be due to a lack of statistical power provided by the modest sample size of 31 subjects (*Button et al., 2013*; *Turner et al., 2018*). Notably, however, previous studies (e.g. *Pessiglione et al., 2006* used a sub-stantially smaller sample size, and in our design power was increased due to the within-subjects design. Second, L-dopa has a plasma half-life of only 60 to 90 min and reaches peak plasma concen-trations (tmax) about 30 to 60 min after oral ingestion (*Baruzzi et al., 1987*; *Keller et al., 2011*; *Iwaki et al., 2015*). However, the time schedule of the current experiment was adjusted to this tmax. In addition, such considerations fall short in explaining the clear behavioral effect of L-Dopa that was observed in the present study.

Obviously, the BOLD signal does not directly reflect DA release, and the precise physiological relationship between DA release and BOLD signal is still to be revealed (*Knutson and Gibbs, 2007*; *Brocka et al., 2018*). A recent optogenetic study in rats suggests that canonical BOLD responses in the reward system may mainly represent the activity of non-dopaminergic neurons, such as glutama-tergic projecting neurons (*Brocka et al., 2018*). Thus, it is also conceivable that L-dopa might have enhanced striatal DA release to some degree without triggering a (detectable) BOLD signal change.

For the haloperidol condition, the null findings on the neural level are less surprising, given the lack of a consistent behavioral effect across subjects. As discussed above, it can be assumed that the

low dose (2 mg) of haloperidol used in this study exerted a mixture of presynaptic (DA-stimulating) and postsynaptic (DA-antagonizing) effects across subjects, potentially explaining why no overall haloperidol effects were found on the behavioral and neural level. Similarly, *Pine et al., 2010* also did not observe significant effects of haloperidol (1.5 mg) on reward-related striatal activity or choice behavior. Future studies should consider using higher doses of haloperidol to achieve more consistent antidopaminergic effects from postsynaptic D2 receptor blockade across subjects, or other DA antagonists with a lower side effect profile.

## L-dopa attenuates neural tracking of overall uncertainty

We reasoned that L-dopa may have reduced exploration not by directly affecting the neural signatures of explore/exploit decisions, but instead by modulating the neural correlates involved in behavioral switching from exploitation to exploration in response to accumulating uncertainty. Thus, L-dopa might delay the time point at which directed exploration is triggered, resulting in less directed exploration trials over time.

We examined this alternative hypothesis with an additional model-based fMRI analysis, in which a trial-by-trial estimate for overall uncertainty (summed standard deviation over all bandits), was used as a parametric regressor in the GLM. Activity in the bilateral insula and dACC positively correlated with the overall uncertainty in the placebo condition, suggesting that these regions may either track the overall uncertainty directly or encode an affective or motivational state that increases with accumulating uncertainty. Insula and dACC thus may trigger exploration under conditions of high overall uncertainty, for example by facilitating switching between the currently exploited option and salient, more uncertain choice alternatives (*Kolling et al., 2012*, p.97; *Laureiro-Martínez et al., 2015*). Indeed, numerous studies have found greater activation in these regions during decision-making under uncertainty, and have implicated both regions in encoding outcome uncertainty or risk (*Huettel et al., 2005*; *Preuschoff et al., 2006*; *Preuschoff et al., 2008*; *Christopoulos et al., 2009*; *Singer et al., 2009*; *Bach and Dolan, 2012*; *Dreher, 2013*). The insula is also considered to play a key role for integrating interoceptive signals about bodily states into conscious feelings, such as urgency, that can influence decision-making under risk and uncertainty (*Craig, 2002*; *Craig, 2009*; *Critchley, 2005*; *Naqvi and Bechara, 2009*; *Singer et al., 2009*; *Xue et al., 2010*). The ACC is known to monitor response conflict, which should increase with the overall uncertainty, and to trigger attentional and behavioral changes for reducing future conflict (*Botvinick et al., 2004*; *Kerns et al., 2004*; *Vanveen and Carter, 2002*).

Importantly, we found that L-dopa reduced uncertainty-related activity in the insula and dACC compared to placebo. Expression of D1 and D2 receptors is much higher in striatum than in insula and ACC (*Hall et al., 1994*; *Hurd et al., 2001*). Hence, L-dopa may have affected uncertainty-related activity in the insula and ACC indirectly by modulating DA transmission on the striatal level. More specifically, L-dopa might have modulated striatal processing of reward uncertainty (*Preuschoff et al., 2006*; *Schultz et al., 2008*) that subsequently is transmitted to cortical structures for integration with other decision parameters to guide explore/exploit behavior (*Kennerley et al., 2006*; *Rushworth and Behrens, 2008*; *Haber and Knutson, 2010*; *Shenhav et al., 2013*). To further test this hypothesis, future studies should more closely examine the role of the insula and ACC in triggering exploration in response to accumulating uncertainty and further investigate how frontal and/or striatal DA transmission might modulate this process.

## Limitations

In addition to the limitations discussed above, the moderate sample size of 31 subjects in this within-subjects manipulation study may have contributed to the absence of haloperidol effects and the absence of drug-associated differences in categorical contrasts of explore/exploit trials in the fMRI analysis. A second limitation relates to the fact that while the applied pharmacological fMRI approach can examine DA drug effects on the BOLD signal, it remains unclear which effects directly reflect local changes in DA signaling, and which reflect downstream effects that may also involve other neurotransmitter systems (*Schrantee and Reneman, 2014*). Needless to say, the BOLD signal provides an indirect index of blood oxygenation rather than a direct measure of DA activity. Hence, an observed BOLD signal change must not necessarily rely on a change in DA transmission, and a change in DA transmission must not necessarily produce a (detectable) BOLD signal change

(*Brocka et al., 2018*). Future research should therefore complement pharmacological fMRI studies with other in vivo techniques that specifically monitor local changes in DA activity, such as molecular imaging with PET and SPECT (single photon emission computed tomography) in humans (*Cropley et al., 2006*).

## Conclusion

The present study examined the causal role of DA in human explore/exploit behavior in a pharmacological model-based fMRI approach, using the dopamine precursor L-dopa and the D2 antagonist haloperidol in a placebo-controlled, within-subjects design. First, our cognitive modeling results confirm that humans use both random and directed exploration to solve the explore/exploit tradeoff. Notably, we extend previous findings by showing that accounting for choice perseveration improves model fit and interpretability of the parameter capturing directed exploration. Our results support the notion that DA is causally involved in the explore/exploit trade-off in humans by regulating the extent to which subjects engage in directed exploration. Interestingly, our neuroimaging data do not support the hypothesis that DA controls this trade-off by modulating the neural signatures of exploratory and exploitative decisions per se. In contrast, we provide first evidence that DA modulates tracking of overall uncertainty in a cortical control network comprising the insula and dACC, which might then drive exploration in the face of accumulating uncertainty. Future research should more closely examine the potential role of these regions in driving exploration based on emotional responses to increasing uncertainty, and further investigate how prefrontal and/or striatal DA may be involved in this process.

## Materials and methods

### Participants

In total, 34 healthy male subjects participated in the study (aged 19 to 35 years, M = 26.85, SD = 4.01). Three subjects dropped out of the study due to illness or personal reasons, two after the initial baseline session and one after the first fMRI session. Only males were included, as female sex hormones fluctuate during menstrual cycle which may affect DA signaling (*Almey et al., 2015*; *Yoest et al., 2018*). Sample-size (n = 31, within-subject design) was based on previous work regarding exploration-related brain activation (*Daw et al., 2006*; n = 14), and dopaminergic manipulation of reinforcement learning (*Pessiglione et al., 2006*; n = 13, between-subject design). We aimed at a sample-size of least twice the size of the above-mentioned studies for replication purposes. Participants were recruited online and included mainly university students. Inclusion criteria were the following: male, age 18–35 years, normal weight (BMI 18.5–25.0), right-handed, fluent German in speaking and writing, normal or corrected to normal vision, no hearing impairments, no major past or present psychological, neurological, or physical disorders, non-smoker, no excessive consumption of alcohol (<10 glasses per week), no consumption of illegal drugs or prescription drugs within two months prior to the study, no irreversibly attached metal in or on the body, no claustrophobia (the latter two due to the fMRI measurement). Before participating in the study, all subjects provided informed written consent and had to pass a medical check by a physician including an electrocardiogram (ECG) and an interview about their medical history and present health status. Participants received a fixed amount (270€) plus a variable bonus depending on task performance (30–50€). The study procedures were approved by the local ethics committee (Hamburg Medical Council).

### General procedure

We employed a double-blind, placebo-controlled, counterbalanced, within-subjects design. Each subject (n = 31) was tested in four separate sessions: one baseline session and three fMRI sessions. The baseline screening was scheduled five to six days prior to the first fMRI session.

### Baseline screening

The baseline screening started with spontaneous eye blink rate assessment, followed by a computerized testing of working memory capacity comprising four working memory tasks (as proxy measures for participants' DA baseline; see Appendix 1), two tasks testing delay and probability discounting behavior (not reported here), and it ended with a psychological questionnaire battery. Participants

were encouraged to take small breaks in between the tasks to aid concentration. Blink rate was measured via electromyography for 5 min under resting conditions via three Ag/AgCl electrodes (*Blumenthal et al., 2005*) and an MP100 hardware system running under AcqKnowledge (version 3.9.1; Biopac Systems, Goleta, CA). Working memory capacity was based on the following tasks: Rotation Span (*Foster et al., 2015*), Operation Span (*Foster et al., 2015*), Listening Span (*van den Noort et al., 2008*; based on the English version by *Daneman and Carpenter, 1980*), and Digit Span (Wechsler Adult Intelligence Scale: WAIS-IV; *Wechsler, 2008*). All tasks were implemented using the software MATLAB (R2014b; MathWorks, Natick, MA) with the Psychophysics Toolbox extensions (version 3.0.12; *Brainard, 1997*; *Kleiner et al., 2007*).

At the end of the baseline screening, subjects completed a computer-based questionnaire battery assessing demographics, personality traits, addictive behavior, and various symptoms of psychopathology. Most of the questionnaires were assessed for a separate study and are of no further importance here. Only the Symptom Checklist-90-Revised (SCL-90-R; *Derogatis, 1992*; German version by *Franke, 1995*) was used to exclude subjects with psychiatric symptoms. A cut-off was calculated for each subject and transformed into T values based on a German norm sample of male students (see SCL-90-R manual by Franke, 2000, p.310–329). As instructed in this manual, the screening cut-off was set to $T_{GSI} \geq 63$ or $T \geq 63$ for at least two of the nine subscales, which was reached by none of the participants. Further, the Edinburgh Handedness Inventory (*Oldfield, 1971*) was used to ensure that all participants were right-handed.

## Scanning procedure

In the three fMRI sessions, each participant performed two tasks inside the MRI scanner under the three different drug conditions. The procedure for each scanning session was as follows: Upon arrival (2.5 hr before testing in the MRI scanner), participants received a first pill containing either 2 mg haloperidol or placebo (maize starch). Two hours later, subjects received a second pill containing either Madopar (150 mg L-dopa + 37.5 mg benserazide) or placebo. Over the course of the study, each subject received one dose of Madopar in one session, one dose of haloperidol in another session, and two placebo pills in the remaining session (counterbalanced). Half an hours later, subjects first performed the restless four-armed bandit task, followed by an additional short reinforcement learning task (not reported here) both inside the MRI scanner. Both tasks were trained on a practice version outside the scanner a priori. On the first fMRI session, a structural MR image (T1) was additionally obtained. Each fMRI session ended with a post-fMRI testing outside the scanner, to assess several control variables (see Appendix 1). Throughout each fMRI session, further control variables were assessed at different time points, including physical wellbeing parameters and mood (see Appendix 1). Subjects were not allowed to eat or drink anything but water throughout the fMRI session but were offered a small snack (cereal bar) after testing in the fMRI scanner to aid concentration for the post-fMRI testing.

## Restless four-armed bandit task

The restless four-armed bandit task was adapted from *Daw et al., 2006*. The task included 300 trials, which were separated by short breaks into four blocks of 75 trials. Each trial started with the presentation of four different colored squares ('bandits') representing four choice options (*Figure 1*). The squares were displayed on a screen that was reflected in a head coil mirror inside the fMRI scanner. Participants selected one option using a button box held in their right hand. Subjects had a maximum of 1.5 s to indicate their choice. If no button was pressed during that time, a large red X was displayed 4.2 s in the center of the screen indicating a missed trial with no points earned. If subjects pressed a button before the response deadline the selected bandit was highlighted with a black frame. After a waiting period of 3 s during which three black dots were shown within the chosen bandit, the number of points earned in this trial was displayed within the chosen bandit for 1 s. Subsequently, the bandits disappeared, and a fixation cross remained on screen until the trial ended 6 s after trial onset. This was followed by a jittered inter-trial interval (poisson distribution, mean: 2 s (0–5 s)). At the end of the task, the sum of points earned as well as the monetary payout resulting from these points were displayed on screen. Participants were told in advance that 5% of all points earned would be paid out after the experiment (5 cents per 100 points). The mean payoffs of the four bandits drifted randomly across trials according to a decaying Gaussian random walk. We used

the three instantiations from *Daw et al., 2006* for the three fMRI sessions of the current study. One of these instantiations is shown in *Figure 1*. The order of these three instantiations across fMRI sessions was the same for all subjects, thereby unconfounded with the drug order, which was counterbalanced across subjects. The task was implemented using the software MATLAB (R2014b; MathWorks, Natick, MA) with the Psychophysics Toolbox extensions (version 3.0.12; *Brainard, 1997*; *Kleiner et al., 2007*).

## Computational modeling of explore/exploit behavior

Choice behavior in the four-armed bandit task was modeled using eight different computational models of explore/exploit choice behavior (*Table 1*). The best fitting model (in terms of predictive accuracy) was selected for subsequent analyses of behavioral and fMRI data, and pharmacological intervention effects. Each computational model was composed of two components: First, a learning rule (Delta rule, Bayesian learner) described how participants generate and update subjective reward value estimates for each choice option (bandit) based on previous choices and obtained rewards. Second, a choice rule (softmax, softmax + exploration bonus, softmax + exploration bonus + perseveration bonus, softmax + exploration bonus + (total uncertainty dependent) random exploration bonus + perseveration bonus) modeled how these learned values influence choices. By combining two different learning rules with four different choice rules, a total of eight models entered for model comparison.

For the sake of brevity, here we only outline the architecture of the Bayesian learner models (see Appendix 1 for the models implementing the Delta rule), which consistently outperformed the Delta rule models (*Daw et al., 2006*). This model implements the Kalman filter (*Anderson and Moore, 1979*; *Kalman, 1960*; *Kalman and Bucy, 1961*) as the Bayesian mean-tracking rule for the reward-generating diffusion process in the bandit task. The model assumes that subjects form an internal representation of the true underlying reward structure of the task. The payoff in trial $t$ of bandit $i$ followed a decaying Gaussian random walk with mean payoff $\mu_{i,t}$ and variance $\sigma_o^2 = 4^2$ (observation variance). From one trial to the next, the mean payoffs changed according to: $\mu_{i,t+1} = \lambda \mu_{i,t} + (1-\lambda)\vartheta + v_t$ with parameters $\lambda$ = 0.9836 (decay parameter), $\vartheta$ = 50 (decay center), and diffusion noise $v_t$ drawn independently in each trial from a Gaussian distribution with zero mean and $\sigma_d^2 = 2.8^2$ (diffusion variance). In the model, subjects' estimates of these parameters are denoted accordingly as $\hat{\lambda}$, $\hat{\vartheta}$, $\hat{\sigma}_d^2$ and $\sigma_d^2$. According to the model, participants update their reward expectations of the chosen bandit according to Bayes' theorem. They start each trial with a prior belief about each bandit's mean payoff, that is normally distributed with mean $\mu_{i,t}^{pre}$ and variance $\sigma_{i,t}^{2pre}$ for bandit $i$ on trial $t$. For the chosen bandit, this prior distribution is updated by the reward observation $r_t$, resulting in a posterior distribution with mean $\mu_{i,t}^{post}$ and variance $\sigma_{i,t}^{2post}$ according to:

$$\hat{\mu}_{c_t,t}^{post} = \hat{\mu}_{c_t,t}^{pre} + \kappa_t \delta_t \; with \; \delta_t = r_t - \hat{\mu}_{c_t,t}^{pre}, \qquad (1)$$

$$\hat{\sigma}_{c_t,t}^{2post} = (1 - \kappa_t)\hat{\sigma}_{c_t,t}^{2pre}. \qquad (2)$$

Here, $\kappa$ denotes the Kalman gain that is computed for each trial $t$ as:

$$\kappa_t = \hat{\sigma}_{c_t,t}^{2pre} / \left(\hat{\sigma}_{c_t,t}^{2pre} + \hat{\sigma}_o^2\right). \qquad (3)$$

The Kalman gain determines the fraction of the prediction error that is used for updating. In contrast to the learning rate (Delta rule), the Kalman gain varies from trial to trial depending on the current variance of the expected reward's prior distribution ($\sigma_{c_t,t}^{2pre}$) and the estimated observation variance ($\sigma_o^2$). The observation variance indicates how much the actual rewards vary around the (to be estimated) mean reward of a bandit and therefore reflects how reliable each trial's reward observation (each new data point) is for estimating the true underlying mean. If the prior variance is large compared to the estimated observation variance, that is if a subject's reward prediction is very uncertain while the reward observation is very reliable, the Kalman gain approaches one and a large fraction of the prediction error is used for updating. If, in contrast, the prior variance is very small compared to the estimated observation variance, that is if a subject's reward estimation is very

reliable while reward observations are very noisy, then the Kalman gain approaches 0 and only a small fraction of the prediction error is used for updating. Similar to the Delta rule, the expected rewards (prior mean and variance) of all unchosen bandits are not updated *within* a trial, that is their posteriors equal the prior for that trial. However, prior distributions of all four bandits are updated *between* trials based on the subject's belief about the underlying Gaussian random walk by:

$$\hat{\mu}_{i,t+1}^{\text{pre}} = \hat{\lambda}\hat{\mu}_{i,t}^{\text{post}} + \left(1 - \hat{\lambda}\right)\hat{\vartheta} \text{ and } \hat{\sigma}_{i,t+1}^{2\text{pre}} = \hat{\lambda}^2\hat{\sigma}_{i,t}^{\text{post}} + \hat{\sigma}_{\text{d}}^2. \tag{4}$$

The trial-by-trial updating process was initialized for all bandits with the same prior distribution $N\left(\mu_1^{pre}, \sigma_1^{2pre}\right)$, with $\mu_1^{pre}$ and $\sigma_1^{2pre}$ as additional free parameters of the model.

The three choice rules were based on the softmax function (*Sutton and Barto, 1998*; *McFadden, 1974*). The first implementation utilized the softmax (SM) with the inverse temperature parameter β modeling inherent choice randomness (random exploration). The second choice rule (SM+E) modeled directed exploration in addition, the third choice rule further modeled choice perseveration (SM+E+P), and the fourth model additionally accounted for total uncertainty-based random exploration (SM+E+R+P), all three via a bonus that was added to the expected value. The resulting probabilities $P_{i,t}$ to choose bandit $i$ on trial $t$ were then:

$$\text{Choice rule 1 (SM) : } P_{i,t} = \frac{\exp(\beta\hat{\mu}_{i,t}^{\text{pre}})}{\sum_j \exp(\beta\hat{\mu}_{j,t}^{\text{pre}})}, \tag{5}$$

$$\text{Choice rule 2 (SM + E) : } P_{i,t} = \frac{\exp(\beta\hat{\mu}_{i,t}^{\text{pre}} + \varphi\hat{\sigma}_{i,t}^{\text{pre}}])}{\sum_j \exp(\beta[\hat{\mu}_{j,t}^{\text{pre}} + \varphi\hat{\sigma}_{j,t}^{\text{pre}}])}, \tag{6}$$

$$\text{Choice rule 3 (SM + E + P) : } P_{i,t} = \frac{\exp(\beta\hat{\mu}_{i,t}^{\text{pre}} + \varphi\hat{\sigma}_{i,t}^{\text{pre}} + I_{c_{t-1=i}}\rho])}{\sum_j \exp(\beta[\hat{\mu}_{j,t}^{\text{pre}} + \varphi\hat{\sigma}_{j,t}^{\text{pre}} + I_{c_{t-1=j}}\rho])}. \tag{7}$$

$$\text{Choice rule 4 (SM + E + R + P) : } P_{i,t} = \frac{\exp(\beta[\hat{\mu}_{i,t}^{\text{pre}} + \varphi\hat{\sigma}_{i,t}^{\text{pre}} + I_{c_{t-1=i}}\rho + \gamma\frac{\hat{\mu}_{i,t}^{\text{pre}}}{\Sigma\hat{\sigma}^{\text{pre}}}])}{\sum_j \exp(\beta[\hat{\mu}_{j,t}^{\text{pre}} + \varphi\hat{\sigma}_{j,t}^{\text{pre}} + I_{c_{t-1=j}}\rho + \gamma\frac{\hat{\mu}_{i,t}^{\text{pre}}}{\Sigma\hat{\sigma}^{\text{pre}}}])}. \tag{8}$$

Choice rule 2 is the "softmax (random exploration) with exploration bonus (directed exploration)", as used in *Daw et al., 2006*. Here, $\varphi$ denotes the exploration bonus parameter, which reflects the degree to which choices are influenced by the uncertainty associated with each bandit.

Choice rule 3 is a novel extension of this model called "softmax with exploration and perseveration bonus". It includes an extra perseveration bonus, which is a constant value (free parameter) only added to the expected value of the bandit chosen in the previous trial. Here, $\rho$ denotes the perseveration bonus parameter and $I$ an indicator function that equals 1 for the bandit that was chosen in the previous trial (indexed by $c_{t-1}$) and 0 for all other bandits.

Choice rule 4 further extends choice rule 3 by adding a second random exploration term that is discounted by the estimated total (i.e. summed) uncertainty of all bandits (*Gershman, 2018*; *Thompson, 1933*). Here, $\gamma$ denotes the uncertainty-based random exploration parameter which captures the recent observation that choice randomness may increase with increasing overall uncertainty (*Gershman and Tzovaras, 2018*).

As mentioned above, all four choice rules were also implemented within a simple Delta rule learning scheme (see Appendix 1). Taken together, by combing each learning rule with each choice rule, eight cognitive models entered model comparison. The parameters for each model are summarized in *Table 1*.

Posterior parameter distributions were estimated for each subject and drug condition using hierarchical Bayesian modeling within Stan (version 2.17.0; *Stan Development Team, 2017*), operating under the general statistical package R (version 3.4.3; *R Development Core Team, 2017*). Stan is based on Hamiltonian Monte Carlo sampling (*Girolami and Calderhead, 2011*) for approximation. Sampling was performed with four chains, each chain running for 1000 iterations without thinning after a warmup period of 1000 iterations. The prior for each group-level mean was uniformly distributed within the limits as given in *Figure 9*. For each group-level standard deviation, a half-Cauchy

distribution with location parameter 0 and scale parameter 1 was used as a weakly informative prior (*Gelman, 2006*). Priors for all subject-level parameters were normally distributed with a parameter-specific mean and standard deviation (denoted by $M^x$ and $\Lambda^x$ for any parameter $x$). Group-level posterior distributions of the three parameters ($\beta, \varphi, \rho$; mean and standard deviation) were estimated separately for each drug condition, which allowed the comparison of subject-level as well as group-level parameters between drugs (for details on the fixed parameters see Appendix 1).

## Model comparison

Following parameter estimation, the eight cognitive models were compared in terms of predictive accuracy using a Bayesian leave-one-out (LOO) cross-validation approach (*Vehtari et al., 2017*). LOO cross-validation computes pointwise out-of-sample predictive accuracy by repeatedly taking one data set ('testing set') out of the sample, refitting the model to the reduced data ('training set'), and then measuring how accurately the refitted model predicts the data of the testing set. A testing set was defined as the data of one subject under one drug condition, compounded over all trials. Model comparison was performed using the data sets from all 31 participants once combined over all drug conditions (yielding 93 data sets) and once separately for each drug condition (each with 31 data sets). To reduce computational burden, the R package loo (*Vehtari et al., 2017*) was used, which applies Pareto-smoothed importance sampling to calculate LOO estimates as a close approximation. LOO estimates were calculated for each model fit based on its Stan output, using the log likelihood function evaluated at the sampled posterior parameter values. The log likelihood for each subject was calculated as the logarithmized product of choice probabilities ($P$) of the chosen bandits (indexed by $c_t$) compounded over trials: $\log\left(\prod_t P_{c_t,t}\right)$. Please note that since cross-validation measures like LOO are not biased in favor of more complex models (like ordinary goodness-of-fit measures), no penalty term is needed here to compensate for model complexity in order to prevent overfitting. Based on the results of the model comparison (*Figure 3*), the cognitive model with the highest predictive accuracy (Bayes-learner with exploration and perseveration bonus (choice rule 3)) was then selected for further data analysis.

## FMRI data acquisition

Functional imaging data were acquired on a Siemens Trio 3T scanner (Erlangen, Germany) equipped with a 32 channel head-coil. For each subject and drug condition, four blocks à 75 trials were recorded for the bandit task. The first five scans of each block served as dummy scans to allow for magnetic field saturation and were discarded. Functional volumes were recorded using a T2*-weighted EPI sequence. Each volume consisted of 40 slices with 2 mm isotropic voxels and 1 mm gap, acquired with a repetition time of 2470 ms, an echo time of 26 ms, and a flip angle of 80°. In addition, a high-resolution structural image was acquired for each subject at the end of the first fMRI session, using a T1-weighted magnetization prepared rapid gradient echo (MPRAGE) sequence with 1 mm isotropic voxels and 240 slices. The experimental task was projected onto a mirror attached to the head coil and participants responded by using a button box with four buttons held in the right hand.

## FMRI data analysis

### Preprocessing

Preprocessing and statistical analysis of fMRI data was performed using SPM12 (Wellcome Department of Imaging Neuroscience, London, UK). The preprocessing included four steps: (1) realignment and unwarping to the first image of the placebo session; (2) slice time correction to the onset of the middle slice; (3) spatial normalization to Montreal Neurological Institute (MNI) space utilizing the DARTEL approach (Ashburner, 2007) with a resampling of functional images to 1.5 mm isotropic resolution; (4) spatial smoothing using a Gaussian kernel of 6 mm full-width at half-maximum (FWHM).

### First-level analysis

For the first-level analysis of fMRI data, a general linear model (GLM) was created for each subject and drug condition, concatenated over all four blocks of the bandit task. For each trial in which a bandit was chosen, two different time points were included in the model: trial onset and outcome

presentation. GLM regressors for these time points were created by convolving these event onsets (stick function of zero duration) with the canonical hemodynamic response function (HRF). Parametric modulators for both onset regressors were included in the model: (1) the type of each choice (1 = explore, 0 = exploit, see *Daw et al., 2006*) as a parametric modulator for the trial onset regressor; (2) the reward prediction error and the outcome value as separate parametric modulators for the outcome regressor. For trials in which no bandit was chosen, the model contained an additional error regressor. Four sessions constants (not convolved with the HRF) were included in the model. Low-frequency noise was removed by employing a temporal high-pass filter with a cut-off frequency of 1/128 Hz, and a first order autoregressive model AR(1) was used to remove serial correlations. Regressor-specific contrast images were created for each subject and drug condition for the five regressors of interest: trial onset, outcome onset, choice type, prediction error, and outcome value.

In addition to the main GLM, two alternative GLMs were created. Both alternative GLMs only differed from the main GLM with respect to the regressors modeled at trial onset, while the remaining regressors were the same. The second GLM included one trial onset regressor with two parametric modulators: the expected value ($\mu^{pre}$) and uncertainty ($\sigma^{pre}$) of the chosen option (in that order), both derived from the cognitive model as described in the results section. The third GLM included three trial onset regressors: one for directed explorations (directed), one for random explorations (random), and one for exploitations (exploit). These three choice types were defined according to the trinary classification scheme as described in the results section.

## Second-level analysis

Utilizing a second-level random effects analysis approach, the subject- and drug-specific contrast images for each first-level regressor were submitted to a flexible factorial model in SPM12, including the factors drug (three levels, within-subject), subject (31 levels), and a constant. For each contrast-specific second-level analysis, a t-contrast image was created that tested for the main effect of that specific contrast over all subjects and drug conditions, calculated by weighting each drug level by one and each subject level by 3/31 (*Gläscher and Gitelman, 2008*). For the choice type regressor (explore = 1, exploit = 0), t-contrast were computed twice, once with positive and once with negative weights to create t-contrast images for both comparisons explore > exploit and exploit > explore. For the second GLM, t-contrasts for both parameteric modulators, that is expected value and uncertainty, were included in the second-level random effects analysis. For the third GLM, the second-level random effects analysis included the t-contrasts directed > exploit, random > exploit, directed > random, and random > directed.

To test for DA drug effects across subjects, an F-contrast image was created for each contrast-specific second-level analysis with the weights [1 -1 0; 0 1 -1] over the three drug levels [P D H] and zero weights for all 31 subject levels (*Henson and Penny, 2005*).

In addition, a second-level regression analysis was conducted for each drug pair to test whether DA drug effects on exploration-specific brain activity were linearly predicted by DA drug effects on exploratory behavior. For this, the subject- and drug-specific contrast images for explore vs. exploit were used to calculate the difference image of this contrast for a given drug pair (P-D, P-H, or D-H) for each subject. These difference images entered a second-level regression analysis, including the subject-specific drug differences of the exploration bonus parameter $\varphi$ posterior medians for the same drug pair as explanatory variable. The same kind of regression analysis was also performed for the contrasts of expected value and uncertainty of the second GLM, and directed vs. exploit and random vs. exploit of the third GLM.

Post-hoc, a fourth first-level GLM was created for an additional exploratory analysis. This fourth GLM differed from the main GLM only with respect to the parametric modulator of the trial onset regressor, replacing the binary variable choice type (explore/exploit) by a continuous model-based variable termed overall uncertainty ($\Sigma\sigma^{pre}$), which is the summed uncertainty ($\sigma^{pre}$) over all four bandits. The contrast images for the overall uncertainty regressor were then used in a second-level random effects analysis to test for drug differences in the parametric effects of this regressor across subjects. Since this post-hoc analysis specifically focused on a comparison of the placebo and L-dopa condition (based on the behavioral findings), the second-level model only included these two drug conditions. Based on this model, different t-contrast images were created to test for the

parametric effects of this regressor in the placebo condition alone, and for its differential parametric effects between both drug conditions (placebo >L dopa, L-dopa >placebo).

For completeness, also a second-level analysis with all three drug conditions was performed to test for the remaining pairwise drug effects accordingly (placebo > haloperidol, haloperidol > placebo, L-dopa > haloperidol, haloperidol > placebo). Finally, also a second-level regression analysis was performed for this regressor.

All fMRI results are reported at a threshold of p<0.05, FWE-corrected for the whole brain volume, unless stated otherwise. In addition, results of the drug-effect related second-level ANOVAs and associated regression analyses were also analyzed using small volume FWE correction (p<0.05) for seven regions that have previously been associated with exploratory choices: the left/right FPC and left/right IPS (*Daw et al., 2006*), as well as the dACC and left/right AI (*Blanchard and Gershman, 2018*). Regions used for small volume correction were defined by a 10 mm radius sphere around the respective peak voxel reported by the previous studies (*Appendix 1—table 5*). For display purposes, an uncorrected threshold of p<0.001 was used (unless stated otherwise), and activation maps were overlaid on the mean structural scan of all participants.

## Acknowledgements

This work was funded by Deutsche Forschungsgemeinschaft (grant PE1627/5-1 to JP). Large parts of this publication are based on the dissertation 'Dopaminergic modulation of the explore/exploit trade-off in human decision making' by KC (Chakroun, Karima: Dopaminergic modulation of the explore/exploit trade-off in human decision making [online]; Hamburg, Univ., Diss., 2019; URL: http://ediss.sub.uni-hamburg.de/volltexte/2019/9835/).

## Additional information

### Funding

| Funder | Grant reference number | Author |
|---|---|---|
| Deutsche Forschungsge-meinschaft | PE1627/5-1 | Jan Peters |

The funders had no role in study design, data collection and interpretation, or the decision to submit the work for publication.

### Author contributions

Karima Chakroun, Conceptualization, Data curation, Formal analysis, Validation, Investigation, Visualization, Methodology, Writing - original draft, Writing - review and editing; David Mathar, Data curation, Formal analysis, Investigation, Visualization, Methodology, Writing - original draft, Writing - review and editing; Antonius Wiehler, Conceptualization, Formal analysis, Methodology, Writing - review and editing; Florian Ganzer, Investigation, Writing - review and editing; Jan Peters, Conceptualization, Resources, Supervision, Funding acquisition, Writing - original draft, Project administration, Writing - review and editing

### Author ORCIDs

David Mathar https://orcid.org/0000-0003-3411-7867
Florian Ganzer http://orcid.org/0000-0002-5894-7964
Jan Peters http://orcid.org/0000-0002-0195-5357

### Ethics

Human subjects: The study procedures were approved (reference number: PV4720) by the local ethics committee (Hamburg Medical Council). All participants provided written informed consent, and consent to publish before participating.

## Decision letter and Author response
Decision letter https://doi.org/10.7554/eLife.51260.sa1
Author response https://doi.org/10.7554/eLife.51260.sa2

# Additional files
## Supplementary files
- Transparent reporting form

## Data availability
STAN modeling code of our (best fitting) computational model (used for subsequent analyses), processed behavioral data underlying Figure 2 and 5, as well as fMRI T-maps underlying Figure 7 and 8 are freely shared on the Open Science Framework, via https://osf.io/vzs63/. Raw behavioral choice data is available after request via https://zenodo.org/record/3872973#.XtZI2GgzaUk.

The following datasets were generated:

| Author(s) | Year | Dataset title | Dataset URL | Database and Identifier |
|---|---|---|---|---|
| Chakroun K, Mathar D, Wiehler A, Ganzer F, Peters J | 2020 | Shared data of "Dopaminergic modulation of the exploration/ exploitation trade-off in human decision-making" | https://osf.io/vzs63/ | Open Science Framework, vzs63 |
| Chakroun K, Mathar D, Wiehler A, Ganzer F, Peters J | 2020 | Dopaminergic modulation of the exploration/exploitation trade-off in human decision-making | https://doi.org/10.5281/ zenodo.3872973 | Zenodo, 10.5281/ zenodo.3872973 |

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

# Appendix 1

## Introduction

To examine potential individual differences in dopaminergic drug effects we also collected putative proxy measures of DA function, namely spontaneous eye blink-rate (sEBR) (*Jongkees and Colzato, 2016*) and working memory capacity (WMC) (*Cools and D'Esposito, 2011*) in a separate 'baseline-session'. We predicted that explore/exploit behavior and their drug-related changes would be modulated by the individual DA baseline, as indexed by our DA proxy measures, according to an inverted-u-shaped function (*Cools and D'Esposito, 2011*; *Cavanagh et al., 2014*).

## Results

### Accounting for perseveration boosts estimates of directed exploration

If perseveration is not explicitly accounted for, variance attributable to perseveration might influence the directed exploration parameter () in the form of an uncertainty-avoiding choice bias (*Badre et al., 2012*; *Payzan-Lenestour and Bossaerts, 2012*). Following up on this, we compared φ estimates for the placebo condition between the winning Bayesian learning model and the reduced model without a perseveration term. Subject-level medians for were highly correlated between models ($r_{29}$=.90, p<.001), but were significantly higher for the model including a perseveration term than for the reduced model (mean difference=0.79, paired t-test: $t_{30}$=7.97, p<.001). This was also true for the corresponding group-level mean parameter of (full model: 0.95; reduced model: 0.16). The number of subjects who showed a negative median, reflecting a discouragement rather than an encouragement of directed exploration was also reduced (full model 6/31; reduced model 13/31). Together, this shows that explicitly accounting for perseveration improved sensitivity to detect effects of directed exploration.

### Correspondence between model parameters and fraction of random exploration, directed exploration and exploitation trials

To verify the correspondence between the trinary trial classification (random exploration, directed exploration, exploitation) and the computational modeling, we used Pearson correlations to test for associations between the model parameters (subject-level medians of β, φ, ρ) and the percentage of the three choice types. To increase the sample size for this correlation, we combined data from the placebo condition (n=31) with data from a prior pilot study using the same task (n=16). With the original binary classification (*Daw et al., 2006*), the percentage of exploration trials per subject was negatively correlated with the random exploration parameter β, but positively correlated with the directed exploration parameter (*Appendix 1—table 1*). However, with the trinary classification, only the percentage of random explorations correlated with β, whereas directed explorations were significantly associated with (*Appendix 1—table 1*), indicating that both parameters indeed reflect different types of exploration.

**Appendix 1—table 1.** Correspondence between model parameters and fraction of random exploration, directed exploration and exploitation trials.

| % explorations | β | φ | ρ |
| --- | --- | --- | --- |
| overall | -.65*** | .30* | 0.18 |
| random | -.68*** | 0.09 | -.22 |
| directed | 0.28 | .64*** | 0.09 |

Note that overall explorations were defined according to the binary choice classification, while directed and random explorations were defined according to the trinary choice classification. $\beta$: RE parameter; $\varphi$: DE parameter; $\rho$: CP parameter. *p<0.05. ***p<0.001.

We checked for drug-related changes on several control variables in addition to the results reported in the main section. The first set of control variables was measured during the post-fMRI testing and included the spontaneous eye blink rate (sEBR), the total scores of the Digit Span Task (forward and backward), and 15 attentional performance measures from the Tests of Attentional Performance (TAP). To test for DA drug effects, a rmANOVA with the factor drug was performed on each of these 18 control variables. None of these 18 variables showed a significant drug effect in the ANOVA (all p>0.05).

A second set of control variables was measured at different time points throughout each fMRI session and comprised six variables on subjective mood (alertness, contentedness, calmness, pleasure, arousal, and dominance) and four variables on physical wellbeing (pulse, systolic and diastolic blood pressure, and the side effects sum score). To test for drug effects on these variables, their scores at three different time points after drug administration (t1, t2, t3) were subtracted by their baseline score before drug administration (t0) for each drug condition. A rmANOVA with the factor drug on each of these difference scores (t1-t0, t2-t0, t3-t0) showed no significant drug effect for any of the physical wellbeing parameters (all p>0.05). However, a significant drug effect ($F_{2,60}$=4.46, p=0.016) was found for one of the difference scores (t3-t0) on the subscale 'calmness' of the VAS (**Bond and Lader, 1974**). Paired t-tests on this variable revealed a significant increase under haloperidol compared to placebo (mean difference P-H = −0.67, t30 = −2.05, p=0.049) and L-dopa (mean difference D-H = −0.85, t30 = −2.99, p=0.005), but no significant difference between placebo and L-dopa (mean difference P-D = 0.18, t30 = 0.61, p=0.544). Although, with correction for multiple comparisons (in total n = 42 tests), the effects no longer remained significant.

Beyond, to test whether subjects were actually blind to the drug condition, their drug guesses after each session were examined. Over all subjects and drug sessions, 30 of the 93 guesses (32.3%) were correct, which is in line with the number of correct guesses expected by chance (31, that is 33.3%). To rule out that some subjects performed above chance (i.e. recognized all drug conditions) and others below, the number of correct guesses per subject was calculated, resulting in the following frequency distribution: 25.8% (0 correct guesses), 51.6% (1 correct guess), 22.6% (2 correct guesses), 0.0% (3 correct guesses). A chi-squared test revealed no significant difference to random guessing (29.6% (0 correct guesses), 44.4% (1 correct guess), 22.2% (2 correct guesses), 3.7% (3 correct guesses); $\chi^2$=1.66, p=0.634; with Monte Carlo approximation). Finally, a chi-squared test showed that the frequencies of drug guesses did not significantly differ between the three drug conditions ($\chi^2_4$=0.36, p=0.986). Taken together, the results of all three analyses indicate that the observed data are in accordance with random guessing.

## Drug effects on the posterior distributions of the single-subject and group-level model parameters

In addition to the results reported in the main results section, we here show the drug-specific posterior distributions of the group-level standard deviations of all free model parameters ($\beta, \varphi, \rho$; **Appendix 1—figure 1a** and **Appendix 1—table 2**), the pairwise drug-related differences of the group-level mean and standard deviation of $\varphi$ (**Appendix 1—figure 1b, c**), and the posterior distributions of the single-subject model parameter $\varphi$ (**Appendix 1—figure 2**). The drug-related differences of the group-level mean and standard deviation for parameter $\varphi$ are also depicted in **Appendix 1—table 2**.

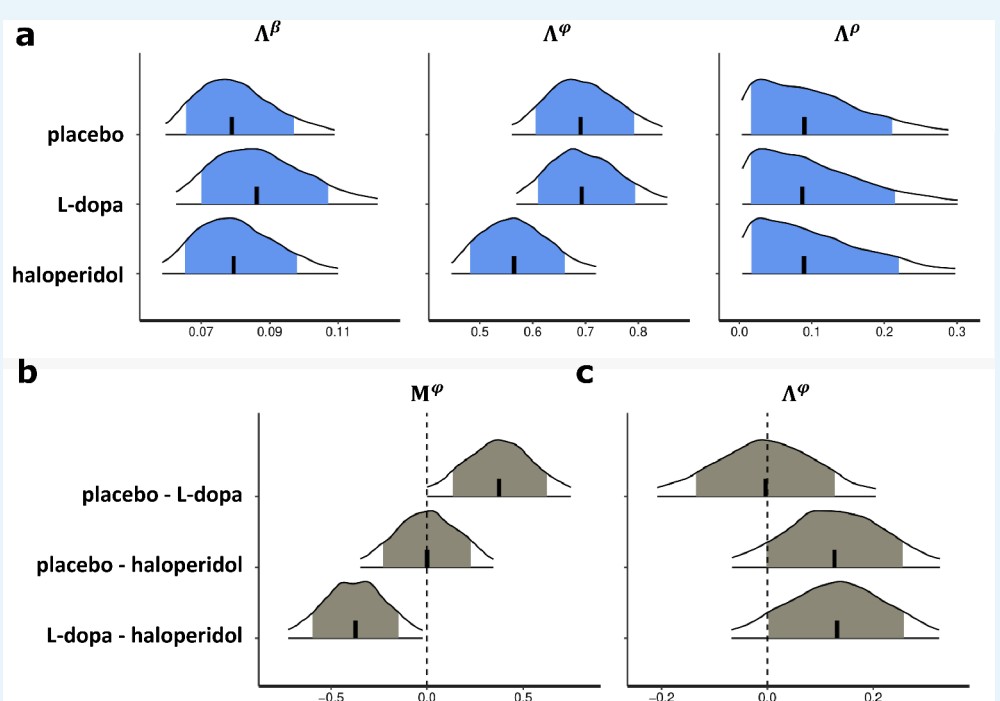

**Appendix 1—figure 1.** Group-level parameter estimates of the winning model . Shown are the posterior distributions of the (**a**) group-level standard deviation ($\Lambda$) for all choice parameters ($\beta, \varphi, \rho$) of the winning model, separately for each drug condition, and (**b**) of the pairwise drug-related differences of the group-level mean (**M**) and (**c**) standard deviation ($\Lambda$) of . For each posterior distribution, the plot shows the median (vertical black line), the 80% central interval (blue/grey area), and the 95% central interval (black contours). $\beta$: softmax parameter; $\varphi$: exploration bonus parameter; $\rho$: perseveration bonus parameter.

**Appendix 1—table 2.** Drug effects on the exploration bonus parameter ($\varphi$) on the group-level.

| | $\mathrm{M}^{\varphi}$ | | $\Lambda^{\varphi}$ | |
|---|---|---|---|---|
| | % above 0 | 90% HDI | % above 0 | 90% HDI |
| placebo - L-dopa | 97.5 | [0.05, 0.69] | 47.5 | [−0.18, 0.16] |
| placebo - haloperidol | 49.3 | [−0.30, 0.27] | 90.0 | [−0.04, 0.29] |
| L-dopa - haloperidol | 1.7 | [−0.70,−0.10] | 90.8 | [−0.02, 0.31] |

Note: Results refer to the posterior drug differences of the group-level mean ($\mathrm{M}^{\varphi}$) and standard deviation ($\Lambda^{\varphi}$) for the $\varphi$ parameter of the winning model. For each posterior difference, the table shows the percentage of samples with values above zero (column: % above 0) and the 90% highest density interval (column: 90%HDI).

In line with the attenuated standard deviation of the group-level $\varphi$ medians, haloperidol reduced the variability of the subject-level $\varphi$ medians (range=[-0.43, 2.33]; SD=0.64) compared to placebo (range=[-0.95, 2.48]; SD=0.85) and L-dopa (range=[-1.77, 2.00]; SD=0.85). Visual inspection suggested that haloperidol increased $\varphi$ for subjects with a relatively low value under placebo and decreased $\varphi$ for subjects with a relatively high $\varphi$ value under placebo (*Appendix 1—figure 2*). We observed similar results in an additional analysis of drug-effects on the percentage of exploitation and exploration trials (overall, random and directed) per subject (see section below). We found no evidence for drug effects on model-free measures of choice behavior (see section below).

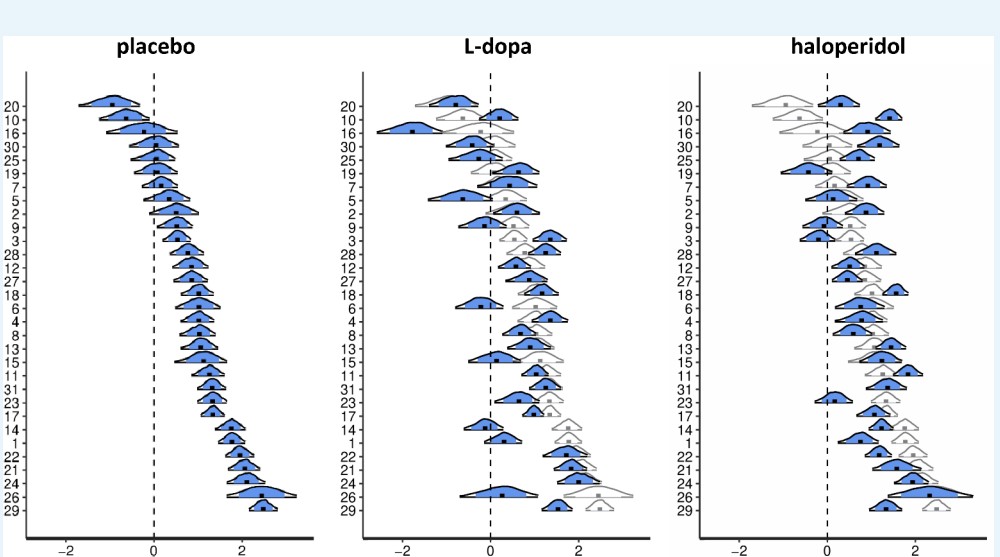

**Appendix 1—figure 2.** Drug effects for the subject-level parameter estimates of the directed exploration parameter $\varphi$. Shown are posterior distributions of the subject-level parameter $\varphi$ from the best-fitting Bayesian model, separately for each drug condition. Each plot shows the median (black dot), the 80% central interval (blue area), and the 95% central interval (black contours). For the L-dopa and haloperidol conditions, posterior distributions (in blue) are overlaid on the posterior distributions of the placebo condition (in white) for better comparison.

## No evidence for the inverted-u-shape dopamine hypothesis

According to the inverted-U-shaped DA hypotheses (*Cools and D'Esposito, 2011*), we predicted participants' individual DA baseline to modulate explore/exploit behavior in a quadratic fashion and to linearly predict the strength and direction of drug-related effects. As potential proxies for subjects' DA baseline, we assessed spontaneous eye blink rate (sEBR; *Jongkees and Colzato, 2016*) and working memory capacity (WMC) via three working memory tasks (Operation/Rotation/Listening SPAN) (*Cools and D'Esposito, 2011*; *Kane et al., 2004*; *Unsworth et al., 2009*; *Redick et al., 2012*). Scores on the first component of a principal component analysis (PCA) (explaining 56.6% of the shared variance) over the z-transformed working memory task scores were used as a working memory compound score (WMC$_{PCA}$). We then used regression to test for linear and quadratic associations between each of the posterior medians of the subject-level model parameters ($\beta$, $\varphi$, $\rho$, *Appendix 1—figure 3*) and the two DA baseline measures (sEBR, WMC$_{PCA}$).

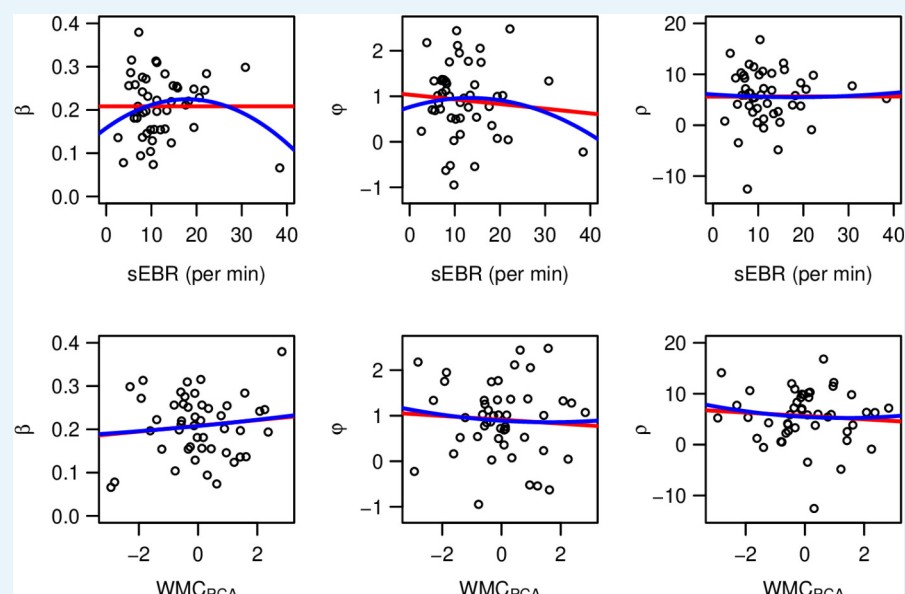

**Appendix 1—figure 3.** Test for an inverted-U relationship between DA baseline proxy measures (spontaneous eye blink rate (sEBR) & working memory capacity (WMC$_{PCA}$)) and the posterior medians of the three choice parameters ($\beta, \varphi, \rho$) of the winning (Bayes-SMEP) model. Model parameters: $\beta$: softmax parameter; $\varphi$: exploration bonus parameter; $\rho$: perseveration bonus parameter.

Models with and without quadratic terms were compared for each proxy measure via LOO cross-validation. In all cases, the fit of the model that included a quadratic term was poorer, and none of the quadratic terms were significant (all p>.05; see **Appendix 1—table 3** for details). We also found no evidence for a linear association between drug-related differences of subject-level model parameters ($\beta, \varphi, \rho$) and DA baseline measures with the help of linear regression models (p>.05 for all regression slopes; **Appendix 1—table 4**).

In addition to the analyses of model-based variables, we also assessed whether drug-associated differences in model-free choice variables were predicted by the individual DA baseline. To increase sample size for this analysis, data from the placebo condition (n = 31) and a pilot study (n = 16) were combined, resulting in a sample of 47 subjects. In summary, we found no evidence for an inverted-U-shaped association of choice behavior and DA proxy measures (**Appendix 1—table 3** and **Appendix 1—figure 4**).

**Appendix 1—table 3.** Test for an inverted-U relationship between choice behavior and DA baseline.

|  | Loo$_{LM}$ - Loo$_{QM}$ | | $\beta_2$ estimate | | $\beta_2$p-value | |
| --- | --- | --- | --- | --- | --- | --- |
|  | **sEBR** | **WMC** | **sEBR** | **WMC** | **sEBR** | **WMC** |
| model-based: | | | | | | |
| $\beta$ | −0.06 | −0.04 | −2.09e$^{-04}$ | 2.98e$^{-04}$ | .132 | .949 |
| $\varphi$ | −3.60 | −2.57 | −1.13e$^{-03}$ | 1.27e$^{-02}$ | .470 | .809 |
| $\rho$ | −53.09 | −49.68 | 1.69e$^{-03}$ | 1.20e$^{-01}$ | .869 | .726 |
| model-free: | | | | | | |
| payout | −0.95 | −1.05 | −6.04e$^{-04}$ | 1.37e$^{-02}$ | .582 | .710 |
| %bestbandit | 198.06 | −245.78 | −2.45e$^{-02}$ | 7.40e$^{-02}$ | .149 | .897 |
| meanrank | 0.06 | −0.10 | −5.29e$^{-04}$ | −3.45e$^{-03}$ | .080 | .733 |
| %switches | −484.04 | −700.67 | −2.09e$^{-04}$ | 6.58e$^{-01}$ | .222 | .509 |

Note. Choice behavior was assessed by the three choice parameters of the winning (Bayes-SM+E +P) model (upper part) and four model-free choice variables (lower part). Baseline dopamine (DA) function was assessed by the two behavioral DA proxies spontaneous eye blink rate (sEBR) and working memory capacity (WMC). For the latter, the first principal component across three different WMC tasks was used, denoted by $WMC_{PCA}$. The column "$LOO_{LM}$-$LOO_{QM}$" denotes the difference of the squared distances for the linear model (LM) minus the quadratic model (QM) from the leave-one-out (LOO) model comparison. Note that negative values for $LOO_{LM}$ - $LOO_{QM}$ indicate better predictive accuracy of the LM. The columns "$\beta_2$ estimate" and "$\beta_2$ p-value" show for each quadratic model the estimated value and p-value of the $\beta_2$ regression coefficient, respectively. Note that data from a pilot study (n=16) and the placebo condition of the main study were combined for this analysis to increase the sample size to n=47. $\beta$: softmax parameter; $\varphi$: exploration bonus parameter; $\rho$: perseveration bonus parameter.

**Appendix 1—table 4.** Test for a linear relationship between drug-related effects on model-parameters and DA baseline.

| | $R^2$ | | $\beta_1$ estimate | | $\beta_1$ p-value | |
|---|---|---|---|---|---|---|
| | **sEBR** | **WMC** | **sEBR** | **WMC** | **sEBR** | **WMC** |
| $\beta$ (P-D) | $2.13e^{-5}$ | $1.52e^{-3}$ | $4.75e^{-05}$ | $2.25e^{-03}$ | .98 | .84 |
| $\varphi$ (P-D) | $1.87e^{-2}$ | $4.25e^{-2}$ | $1.40e^{-02}$ | $-1.19e^{-01}$ | .46 | .27 |
| $\rho$ (P-D) | $5.91e^{-3}$ | $2.42e^{-3}$ | $-4.89e^{-03}$ | $-1.76e^{-01}$ | .68 | .79 |
| $\beta$ (P-H) | $2.47e^{-2}$ | $2.93e^{-2}$ | $1.64e^{-03}$ | $1.01e^{-02}$ | .40 | .36 |
| $\varphi$ (P-H) | $9.58e^{-3}$ | $3.36e^{-2}$ | $-1.01e^{-02}$ | $-1.06e^{-01}$ | .60 | .32 |
| $\rho$ (P-H) | $4.61e^{-2}$ | $1.18e^{-2}$ | $-9.01e^{-03}$ | $-2.57e^{-01}$ | .25 | .56 |
| $\beta$ (D-H) | $1.57e^{-3}$ | $7.83e^{-3}$ | $1.57e^{-03}$ | $7.83e^{-03}$ | .43 | .49 |
| $\varphi$ (D-H) | $1.02e^{-3}$ | $2.95e^{-4}$ | $-3.93e^{-03}$ | $1.20e^{-02}$ | .86 | .93 |
| $\rho$ (D-H) | $4.11e^{-3}$ | $5.69e^{-4}$ | $-4.07e^{-02}$ | $-8.54e^{-02}$ | .73 | .90 |

Note. Drug-related differences (P: placebo, D: L-dopa, H: haloperidol) of model parameters for all participants (n = 31). Baseline dopamine (DA) function was assessed by the two behavioral DA proxies spontaneous eye blink rate (sEBR) and working memory capacity (WMC). For the latter, the first principal component across three different WMC tasks was used, denoted by WMCPCA. The column '$R^2$' denotes the $R^2$-values of the linear regressions. The columns '$\beta_1$ estimate' and '$\beta_1$ p-value' show for each linear model the estimated value and p-value of the $\beta_1$ regression coefficient, respectively. $\beta$: softmax parameter; $\varphi$: exploration bonus parameter; $\rho$: perseveration bonus parameter

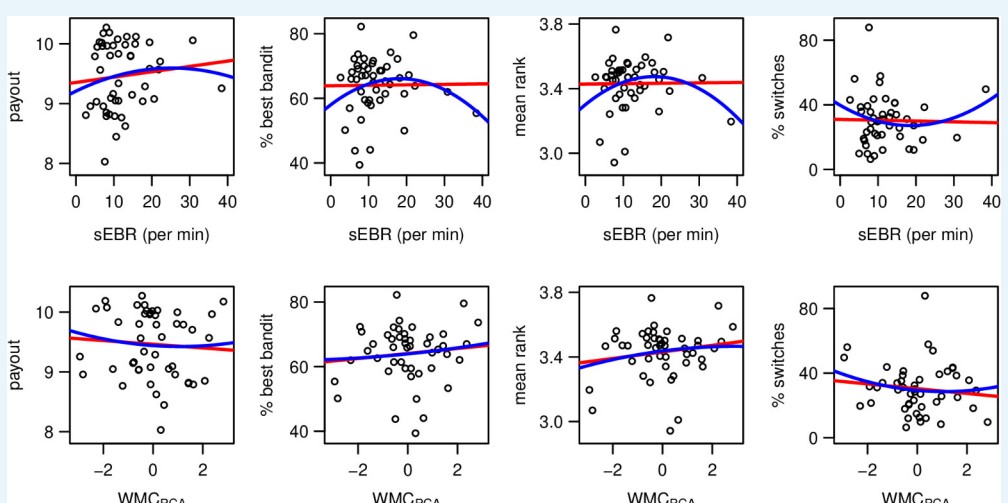

**Appendix 1—figure 4.** Test for an inverted-U relationship between choice behavior and DA baseline. Choice behavior was assessed by four model-free choice variables (*payout, % bestbandit, meanrank, %switches*). DA baseline function was assessed by the two DA proxies spontaneous eyeblink rate (sEBR) and working memory capacity (WMC). For the latter, the first principal component across three different WMC tasks was used, denoted by WMC$_{PCA}$. Each plot shows two regression lines that were fitted to the data, one for the "linear model" (red line) and one for the "quadratic model" (blue line). Note that data from a pilot study and the placebo condition of the main study were combined for this analysis to increase the sample size to n=47. $\beta$: softmax parameter; $\varphi$: exploration bonus parameter; $\rho$: perseveration bonus parameter.

**Appendix 1—table 5.** Regions used for small volume correction.

| region of small volume correction | peak voxel (mm) | | | reference for peak voxel |
|---|---|---|---|---|
| | x | y | z | |
| rFPC (right frontopolar cortex) | 27 | 57 | 6 | *Daw et al., 2006* |
| lFPC (left frontopolar cortex) | −27 | 48 | 4 | *Daw et al., 2006* |
| rIPS (right intraparietal sulcus) | 39 | −36 | 42 | *Daw et al., 2006* |
| lIPS (left intrapareital sulcus) | −29 | −33 | 45 | *Daw et al., 2006* |
| rAIns (right anterior insula) | 32 | 22 | -8 | *Blanchard and Gershman, 2018* |
| lAIns (left anterior insula) | −30 | 16 | -8 | *Blanchard and Gershman, 2018* |
| dACC (dorsal anterior cingulate cortex) | 8 | 16 | 46 | *Blanchard and Gershman, 2018* |

Note: Each small volume correction used a 10-mm-radius sphere around the listed voxel coordinates, which mark brain regions that have previously been associated with exploratory choices.

The center coordinates of the 10 mm spheres used for the regions of interest analysis are reported below (*Appendix 1—table 5*).

Appendix 1—table 6 and *Appendix 1—figure 5a* depict the brain regions that showed higher activity in exploratory compared with exploitative choices as revealed by the first GLM. In *Appendix 1—table 7* and *Appendix 1—figure 5b* the opposite contrast is depicted.

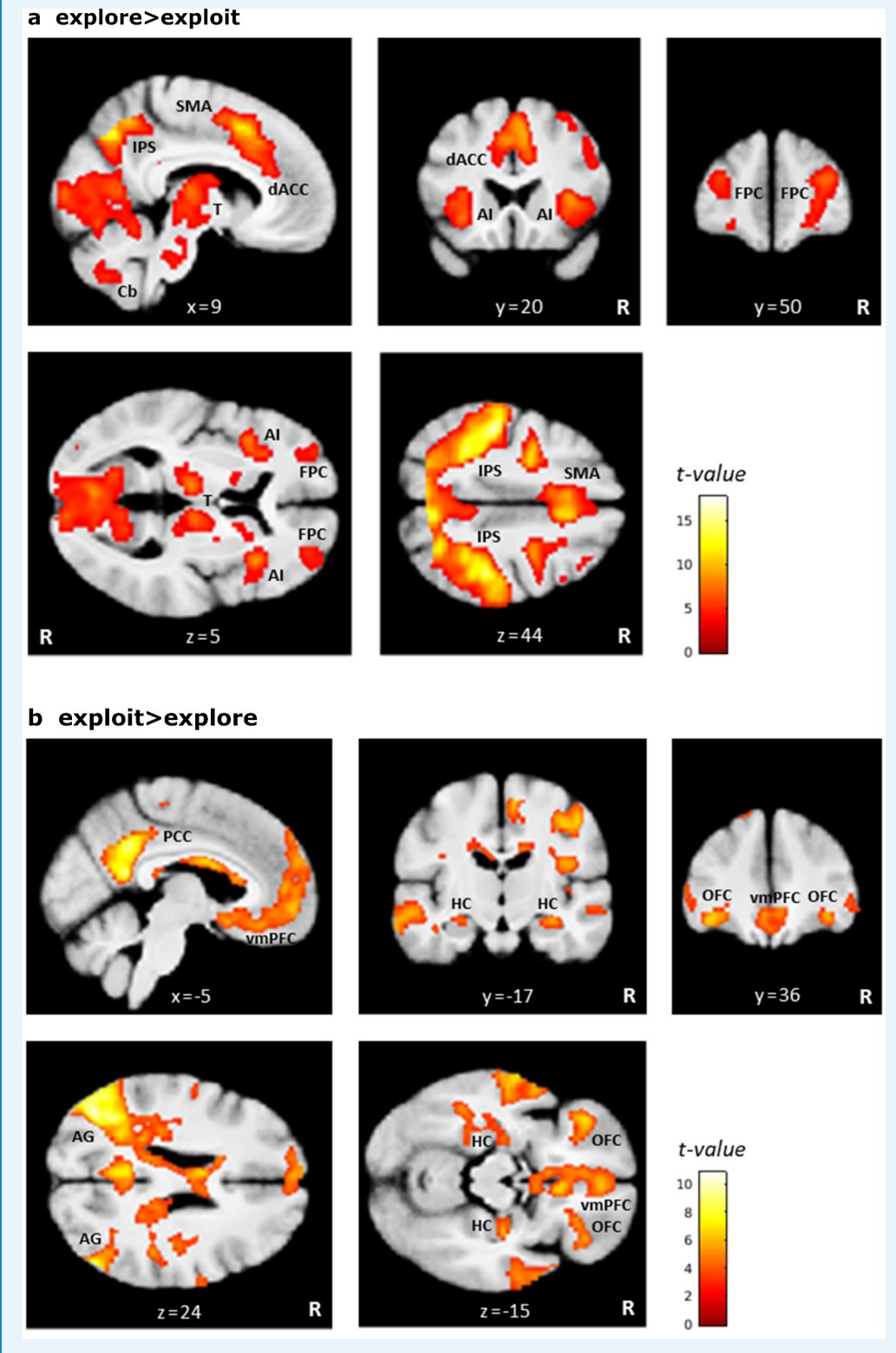

**Appendix 1—figure 5.** Brain regions differentially activated by exploratory and exploitative choices. Shown are statistical parametric maps (SPMs) for (**a**) the contrast explore > exploit and (**b**) the contrast exploit > explore over all drug conditions. AG: angular gyrus; AI: anterior insula; Cb: cerebellum; dACC: dorsal anterior cingulate cortex; FPC: frontopolar cortex; HC: hippocampus; IPS: intraparietal sulcus; vmPFC: ventromedial prefrontal cortex;

OFC: orbitofrontal cortex; PCC: posterior cingulate cortex; SMA: supplementary motor area; T: thalamus. For visualization purposes thresholded at p<0.001, uncorrected. R: right.

**Appendix 1—table 6.** Brain regions showing higher activity for exploratory than exploitative choices (first GLM).

| Region | MNI coordinates | | | peak z-value | cluster extent (k) |
|---|---|---|---|---|---|
| | x | y | z | | |
| R/L intraparietal sulcus, R/L postcentral gyrus, R/L precuneus, L precentral gyrus | −48 | −33 | 52 | 10.45 | 15606 |
| R precentral gyrus | 26 | -8 | 50 | 9.32 | 2297 |
| R/L supplementary motor cortex, R/L dorsal anterior cingulate cortex | 8 | 12 | 45 | 8.47 | 2552 |
| R cerebellum/fusiform gyrus | 18 | −51 | −22 | 8.09 | 2574 |
| R middle frontal gyrus (FPC) | 39 | 34 | 28 | 7.56 | 1291 |
| R cerebellum | 24 | −57 | −54 | 7.35 | 128 |
| L precentral gyrus | −51 | 0 | 34 | 7.31 | 430 |
| L cerebellum, L fusiform gyrus | −40 | −54 | −32 | 7.28 | 1419 |
| L thalamus | −10 | −20 | 6 | 6.96 | 556 |
| R/L calcarine cortex | -8 | −74 | 14 | 6.90 | 1222 |
| R anterior insula | 36 | 20 | 3 | 6.87 | 511 |
| L anterior insula | −36 | 15 | 3 | 6.69 | 557 |
| R precentral gyrus | 51 | 8 | 24 | 6.49 | 434 |
| R thalamus | 10 | −18 | 8 | 6.32 | 331 |
| R cerebellum | 30 | −44 | −48 | 6.24 | 28 |
| L middle frontal gyrus (FPC) | −42 | 27 | 27 | 6.07 | 97 |
| R cerebellum | 14 | −62 | −45 | 5.88 | 61 |
| R pallidum | 15 | 6 | -4 | 5.83 | 25 |
| R calcarine cortex | 9 | −94 | 6 | 5.74 | 104 |
| vermis | 3 | −75 | −34 | 5.70 | 52 |
| R supramarginal gyrus | 51 | −42 | 28 | 5.69 | 46 |
| L middle frontal gyrus (FPC) | −30 | 46 | 15 | 5.67 | 47 |
| L pallidum | −10 | 6 | -4 | 5.64 | 51 |
| R anterior orbital gyrus | 24 | 54 | -9 | 5.60 | 33 |
| L posterior cingulate cortex | -3 | −32 | 26 | 5.51 | 21 |
| L caudate nucleus | −16 | −14 | 18 | 5.33 | 28 |
| R caudate nucleus | 12 | -8 | 16 | 5.24 | 16 |
| L lingual gyrus | −16 | −84 | −12 | 5.21 | 10 |
| R anterior cingulate cortex | 10 | 27 | 21 | 5.13 | 10 |

Note: Thresholded at p<0.05, FWE-corrected for whole-brain volume, with k ≥ 10 voxels; L: left; R: right.

**Appendix 1—table 7.** Brain regions showing higher activity for exploitative than exploratory choices (first GLM).

| Region | MNI coordinates | | | peak | cluster |
|--------|-----|-----|-----|---------|-------------|
| | x | y | z | z-value | extent (k) |
| L angular gyrus | −42 | −74 | 34 | 8.04 | 2530 |
| L posterior cingulate cortex/precuneus | -6 | −52 | 15 | 7.40 | 1087 |
| R angular gyrus | 52 | −68 | 28 | 7.02 | 185 |
| R postcentral gyrus | 33 | −26 | 54 | 6.80 | 503 |
| R cerebellum | 27 | −78 | −38 | 6.28 | 452 |
| R rostral anterior cingulate cortex | 4 | 18 | −14 | 5.90 | 125 |
| L superior temporal gyrus | −62 | −36 | 3 | 5.89 | 70 |
| L lateral orbital gyrus | −38 | 34 | −14 | 5.81 | 102 |
| R central operculum | 45 | −14 | 20 | 5.73 | 83 |
| L middle temporal gyrus | −62 | -4 | −22 | 5.67 | 193 |
| R/L medial frontal cortex (vmPFC) | -2 | 40 | −10 | 5.67 | 233 |
| L superior frontal gyrus | −10 | 54 | 30 | 5.54 | 20 |
| L superior frontal gyrus | −10 | 51 | 36 | 5.45 | 10 |
| L middle temporal gyrus | −60 | −51 | -2 | 5.38 | 61 |
| R superior temporal gyrus | 52 | −12 | -9 | 5.35 | 25 |
| R middle temporal gyrus | 62 | 4 | −21 | 5.30 | 10 |
| L rostral anterior cingulate cortex | -6 | 46 | 4 | 5.17 | 13 |
| L inferior frontal gyrus | −50 | 27 | 2 | 5.16 | 20 |

Note: Thresholded at p<0.05, FWE-corrected for whole-brain volume, with k ≥ 10 voxels; L: left; R: right

In addition to the contrast directed > random exploration trials, we tested for the reverse contrast (random > directed exploration). This yielded a small cluster of three voxels in the right FPC (32, 50,–8 mm; z = 5.34) across conditions. However, the number of trials was highly unequal for both exploration types with on average three times more random than directed exploration trials per session. After exclusion of all sessions with ≤ 5 trials in the directed exploration condition (8 out of 93 sessions), the frontopolar cluster was no longer significant at the whole-brain level. In line, overlaying activation maps for directed, random, and overall explorations (each contrasted against exploitation) showed a highly similar activation pattern for all three exploration conditions (*Appendix 1—figure 6*) that in each case included the same network (bilateral FPC, IPS, dACC, AI, and thalamus).

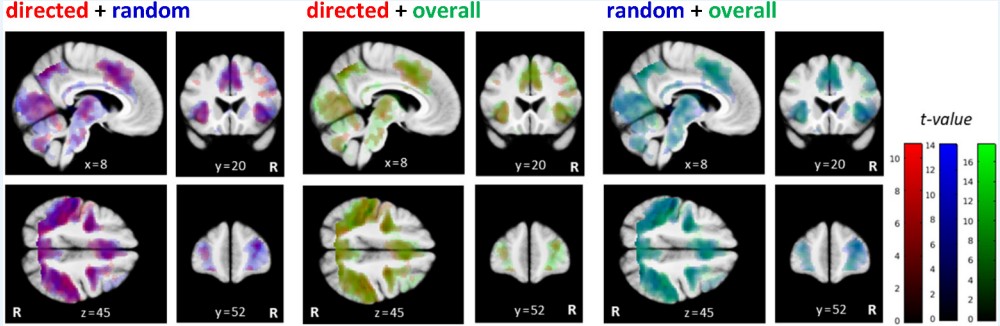

**Appendix 1—figure 6.** Brain activation patterns for different types of explorations . Shown are pairwise overlays of the statistical parametric maps for the contrasts explore > exploit ('overall' in green), directed > exploit ('directed' in red), and random > exploit ('random' in blue) over all drug conditions. While the first contrast is based on a binary choice classification according to which all choices not following the highest expected value are

explorations, the other two contrast are based on a trinary choice classification, which further subdivides explorations into choices following the highest exploration bonus (directed) and choices not following the highest exploration bonus (random). All activation maps thresholded at p<0.05, uncorrected for display purposes. R: right.

## Neural activation in response to model-based prediction errors (PEs)

The model-based PE was positively correlated with activity in bilateral ventral striatum (left: −16, 6,–12 mm; z = 6.40; right: 16, 9, 10 mm; z = 6.20), as shown in *Appendix 1—figure 7*.

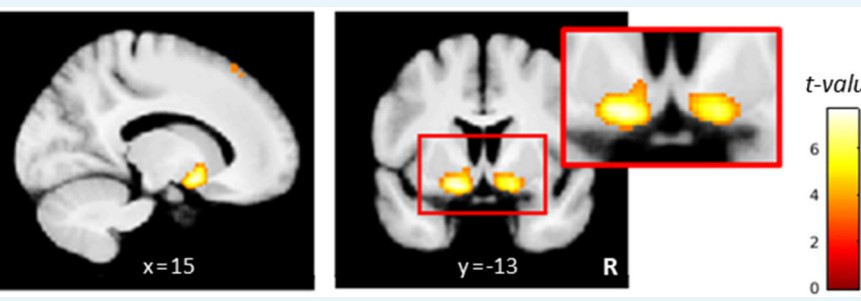

**Appendix 1—figure 7.** Striatal coding of the model-based prediction error (PE). Activity in the bilateral ventral striatum correlated positively with the PE signal. For visualization purposes thresholded at p<0.001, uncorrected. R: right.

## No further drug-related neural activation differences

In addition to the fMRI analyses described in the manuscript, we computed exploratory rmANOVAs for the four remaining regressors of the first GLM (trial onset, reward onset, PE, and outcome value), to explore other drug-related effects on brain activation. These analyses also revealed no suprathreshold activations on the whole-brain level. DA drug effects on directed exploration ($\varphi$) showed considerable variability between subjects both in terms of magnitude and direction (*Appendix 1—figure 2*). Therefore, we also performed second-level regression analyses for each drug pair, testing whether drug effects on exploration specific brain activity (first/second GLM contrasts for explorevs.exploit, directed vs.exploit, and randomvs.exploit) were linearly predicted by the drug effects on exploratory behavior (drug related differences of the subject-specific $\varphi$ medians). However, none of these analyses revealed any suprathreshold effects on the whole brain level, nor in any of the seven a priori ROIs.

# Methods

## Computational modeling (delta rule)

Choice behavior in the four-armed bandit task was modeled using the combination of two different learning rules (Delta learning rule, Bayesian learner), and four choice rules (softmax, softmax + exploration bonus, softmax + exploration bonus + exploitation bonus, softmax + exploration bonus + exploitation bonus + uncertainty-based random exploration bonus) that resulted in eight different computational models of explore/exploit choice behavior. Here, the architecture of the delta learning rule (*Sutton and Barto, 1998*) which is an established temporal difference model of reinforcement learning is outlined.

According to this rule, subjects update the expected reward value ($v$) of a chosen bandit based on their prediction error ($\delta$), that is the difference between the actual reward ($r$) and the expected reward for that trial:

$$v_{c_t, t+1} = v_{c_t, t} + \alpha \delta_t \, \text{with} \, \delta_t = r_t - v_{c_t, t}$$

Herein, the indices $t$ and $t+1$ denote the current and the next trial, respectively, and $c_t$ the index of the bandit chosen on trial $t$. The parameter $\alpha$ denotes the learning rate, which is a free parameter in this model ranging between 0 and 1. The learning rate determines the fraction of the prediction error that is used for updating. In contrast, the expected rewards of all unchosen bandits are not changed from one trial to the next, that is they remain constant until that bandit is chosen again. This trial-by-trial updating was initialized for each bandit with the same expected reward value $v_1 = 50$.

Next, three different choice rules were used to model subjects' choices based on their expected rewards derived from either the Delta rule or the Bayesian learner rule. All choice rules were based on the commonly applied softmax function (**Sutton and Barto, 1998**; **McFadden, 1974**). The first model was the softmax function (SM) in its basic form without any bonus term. According to this rule, choices are probabilistically based on the relative expected reward values of each choice option. The softmax $\beta$ parameter, also called inverse temperature, reflects the degree of randomness (random exploration) in a subject's decisions. The second choice rule was a modified version of the softmax function called "softmax with exploration bonus" (SM+E), adopted from **Daw et al., 2006**. This model adds an additional "exploration bonus" to the expected value of each bandit, which increased with the uncertainty of a bandit's outcome. Within the Delta learning rule models, this was accomplished with a simple heuristic adopted from **Speekenbrink and Konstantinidis, 2015**. According to that heuristic, a bandit's uncertainty increases linearly with the number of trials since it was last chosen. This is formalized as $(t - T_i)$, where $T_i$ is the last trial before the current trial $t$ in which bandit $i$ was chosen. The third choice rule was a novel extension of the second choice model called "softmax with exploration and perseveration bonus" (SM+E +P). This version of the softmax rule includes an extra perseveration bonus, in the form of a constant value (free parameter) only added to the expected value of the bandit chosen in the previous trial, but not for all other bandits. Choice rule 4 (SM+E+R+P) further extended choice rule 3 by adding a random exploration term that is discounted by the estimated total (i.e. summed) uncertainty of all bandits (**Gershman, 2018**; **Thompson, 1933**). This shall capture the recent observation that choice randomness may increase with increasing overall uncertainty (**Gershman and Tzovaras, 2018**).

In analogy to the Bayseian models (see Materials and methods section of the main article), the four resulting models utilizing the Delta learning rule read as follows:

$$\text{Choice rule 1 (SM)} : P_{i,t} = \frac{\exp(\beta v_{i,t})}{\sum_j \exp(\beta v_{j,t})}$$

$$\text{Choice rule 2 (SM + E)} : P_{i,t} = \frac{\exp(\beta[v_{i,t} + \varphi(t - T_i)])}{\sum_j \exp(\beta[v_{j,t} + \varphi(t - T_j)])}$$

$$\text{Choice rule 3 (SM + E + P)} : P_{i,t} = \frac{\exp(\beta[v_{i,t} + \varphi(t - T_i) + I_{c_{t-1=i}}\rho])}{\sum_j \exp(\beta[v_{i,t} + \varphi(t - T_i) + I_{c_{t-1=j}}\rho])}$$

$$\text{Choice rule 4 (SM + E + R + P)} : P_{i,t} = \frac{\exp(\beta[v_{i,t} + \varphi(t - T_i) + I_{c_{t-1=i}}\rho + \gamma\frac{v_{i,t}}{\Sigma(t-T_i)}])}{\sum_j \exp(\beta[\hat{\mu}_{j,t}^{pre} + \varphi\hat{\sigma}_{j,t}^{pre} + I_{c_{t-1=j}}\rho + \gamma\frac{v_{i,t}}{\Sigma(t-T_i)}])}.$$

Herein, $P_{i,t}$ denotes the probability to choose bandit $i$ on trial $t$, and $\sum_j$ indicates a summation over all four bandits; $\varphi$ denotes the exploration bonus parameter, which reflects the degree to which choices are influenced by the uncertainty associated with each bandit, $\rho$ denotes the perseveration bonus parameter, $I$ an indicator function that equals 1 for the bandit that was chosen in the previous trial (indexed by $c_{t-1}$) and 0 for all other bandits, and $\gamma$ denotes the uncertainty-based random exploration bonus.

### Fixed parameters (Bayesian learner)

For all Bayes learner models, the six parameters specifying subjects' estimation of the Gaussian random walk ($\hat{\lambda}$, $\hat{\vartheta}$, $\hat{\sigma}_0^2$, $\hat{\sigma}_d^2$, $\hat{\mu}_1^{pre}$, $\hat{\sigma}_1^{pre}$) ('random walk parameters') were fixed to constrain parameter space for these models, which largely facilitated estimation of the remaining choice parameters. Also, some of the Bayes learner models did actually not converge with free random walk parameters, thus fixing these parameters was necessary in order to include all six cognitive models in the model comparison. In detail, the parameters $\lambda$, $\vartheta$, $\sigma_o^2$, and $\sigma_d^2$ were fixed to the values of the true underlying random walk parameters (see Materials and methods section of the main article). The parameters $\mu_1^{pre}$ and $\sigma_1^{pre}$, specifying the mean and standard deviation of subjects' prior reward expectation for each bandit in the first trial, were fixed to $\mu_1^{pre}$=50 and $\sigma_1^{pre}$=4. Note that while these latter parameter values, which reflect reward expectancies on trial one, were chosen somewhat arbitrarily, they only influence modeled choice behavior on the first few trials and thus have low impact on the overall model fit (see *Daw et al., 2006*).

## Discussion

### Perseveration boosts model fit

It has been argued before that perseveration, that is, repeating choices regardless of value (*Rutledge et al., 2009*; *Schönberg et al., 2007*; *Brough et al., 2008*; *Worthy et al., 2013*) can negatively impact estimates of directed exploration (*Payzan-Lenestour and Bossaerts, 2012*; *Wilson et al., 2014*; *Daw et al., 2006*; *Badre et al., 2012*), leading to smaller or even negative exploration bonus effects. To address this issue, we included a perseveration term in the exploration bonus model (*Daw et al., 2006*). This not only improved model fit, but also substantially increased estimates of directed exploration and reduced the number of subjects showing a negative $\varphi$ estimate.

### No evidence for the inverted-U-hypothesis

We found no evidence for the inverted-U-hypothesis of DA function and cognition. This might be due to several reasons. Despite its popularity, the inverted-U-hypothesis remains relatively vague regarding the specific DA functions and cognitive domains it applies to, and is therefore difficult to test and falsify. Animal studies suggest that it specifically describes the relationship between prefrontal D1 receptor activity and working memory performance, whereas the relation between other aspects of DA action and cognitive functions may follow different functions (*Floresco and Magyar, 2006*; *Floresco, 2013*).

It remains unclear how to construe the exact shape and turning point (optimum) of the inverted-U-shape function, since these aspects may vary across tasks, cognitive functions, and individuals (see *Cools et al., 2008*; *Cools and D'Esposito, 2011*; *Wiegand et al., 2016*). Obviously, our study also has limitations that exacerbate comprehensive testing of this more complex hypothesis. First, the sample size of 31 participants may simply be too small to detect an inverted quadratic relationship between proxies for DA baseline and drug effects on explore/exploit behavior. It has been argued (*Slagter et al., 2012*) that healthy subjects may display only a relatively restricted range in baseline DA levels during resting conditions, making it more difficult to observe inverted-u-shape effects in these samples. Blink rate values were strongly left-skewed across subjects, with only relatively few subjects with high blink rates, supporting this idea. The failure to observe an inverted-U-shaped effect in the current study might also be due to poor DA proxy measures. While the spontaneous blink rate has been extensively investigated as a proxy for DA function in animals and humans, many studies have also yielded conflicting or inconclusive results (*Jongkees and Colzato, 2016*; *Dang et al., 2017*; *Sescousse et al., 2018*). For working memory capacity, the available evidence is even more limited than for the blink rate, as fewer studies have used this measure as a DA proxy. While some studies report differential DA drug effects in relation to individual working memory capacity, the directionality of these effects differs

between studies (*Kimberg et al., 1997*; *Kimberg and D'Esposito, 2003*; *Gibbs and D'Esposito, 2005*). Moreover, for both proxy measures, it is not clear which specific aspect of DA function they might index, and what the underlying mechanism is. Both measures might reflect aspects of striatal DA function, such as striatal D2 receptor availability (*Groman et al., 2014*; *Jongkees and Colzato, 2016*) and/or striatal DA synthesis capacity (*Cools et al., 2008*; *Landau et al., 2009*). In contrast, the inverted-U-shaped hypothesis predominantly relates to D1 receptor function in PFC. In conclusion, the DA proxies used in this study may have failed to validly measure baseline DA function, or specifically measured an aspect of DA function which was not predictive for the behavioral outcome measure under study.

