## [Decision Letter]

**Acceptance summary:**

The reviewers and I agree that this work offers important new data bearing upon the role of dopamine in exploration. We appreciated the thoroughness of the modeling and data analysis.

**Decision letter after peer review:**

Thank you for submitting your article "Dopaminergic modulation of the exploration/exploitation trade-off in human decision-making" for consideration by *eLife*. Your article has been reviewed by Michael Frank as the Senior Editor, a Reviewing Editor, and three reviewers. The following individuals involved in review of your submission have agreed to reveal their identity: Samuel J Gershman (Reviewer #1); Bruno B Averbeck (Reviewer #2).

The reviewers have discussed the reviews with one another and the Reviewing Editor has drafted this decision to help you prepare a revised submission.

Summary:

This paper reports a new drug and neuroimaging dataset using a multi-armed bandit task previously studied by Daw et al., (2006), who also used fMRI (albeit with a smaller sample size). The main new results here come from the drug treatment (L-dopa and haloperidol), since the models studied here are nominally the same (though the authors use a more sophisticated hierarchical Bayesian estimation scheme). Consistent with (some) earlier work, models incorporating both directed exploration (in the form of an exploration bonus tied to bandit arm uncertainty) and perseveration (a tendency to repeat a previous choice) outperform models lacking each of these effects. The main novel finding is that L-dopa reduces the strength of the uncertainty bonus in exploratory choice. While fMRI contrasts between explore and exploit trials did reveal differences in line with previous literature, there were no drug effects that reached statistical significance, save for one exploratory analysis at a reduced threshold. The authors should be commended for comprehensively reporting both positive and null effects. The reviewers also agreed that the paper is for the most part technically well-executed and provides much-needed data on an important set of questions in the reinforcement learning literature.

Essential revisions:

1) The model comparison includes sensible models (it's also nice to see consistent model ordering across conditions). One thing that's missing (as noted in the Discussion section) is a model of random exploration dependent on "total uncertainty" (see Gershman, 2018, 2019), which should theoretically act as a gain modulator of the values. In fact, what's called the "overall uncertainty" is almost identical to this quantity, and is used in later analyses but apparently not in the choice rule.

2) It doesn't quite make sense to link the neural representation of "overall uncertainty" to the uncertainty-based exploration strategy formalized in this paper, because that strategy uses an uncertainty bonus, which means that the critical quantity is *relative* not *overall* (or "total" in the terminology of Gershman, 2018) uncertainty (relative and total uncertainty have been distinguished by previous imaging studies: Badre et al., 2012; Tomov et al., 2019). One approach, as suggested above, would be to build uncertainty-based random exploration into the behavioral model, and then connect the quantities in that model to the neural and drug data. Another (not mutually exclusive) approach is to look for neural correlates of relative uncertainty (i.e., the difference between uncertainties). For example, using the uncertainty of the chosen option relative to the average uncertainty of the unchosen options.

3) The manuscript is quite long and has a lot of detail that could be relegated to supplemental or summarized. Each subtopic received at least a page in the Discussion, comprising one paragraph of discussion of results followed by lengthy literature review and speculation about its relation to other findings. All of the detail is interesting to a small percentage of expert readers. However, for most readers it makes it hard to get at the main results. As many of the outcomes are null findings, this is even more of an issue. For example, Figure 3 could be removed and replaced with some text indicating means and 95% Cis. Figure 4 is too much detail. Also, the paragraph that discusses how the inclusion of the perseverative coefficient in the model increases sensitivity for directed exploration could be in supplemental (this is worth pointing out, since the original Daw study did not find evidence for this, but this will be of interest to a small number of people, especially since several other studies have now found evidence for an uncertainty bonus). Also, the paragraph that shows no evidence for U-shaped dopamine effects and similarly the figure showing RPE effects in the ventral-striatum could go to supplemental. And correspondingly, a bit more behavioral data might be useful. For Figure 2, is there a way to show a summary figure, across subjects? One of the issues with the original Daw study, was that it was very hard for the participants to learn. How well are the participants able to track the values, and find the best option? Is there a way (might be very hard) to illustrate the effect of L-dopa on directed exploration by directly plotting behavioral data?

---

## [Author Response]

Summary:This paper reports a new drug and neuroimaging dataset using a multi-armed bandit task previously studied by Daw et al., (2006), who also used fMRI (albeit with a smaller sample size). The main new results here come from the drug treatment (L-dopa and haloperidol), since the models studied here are nominally the same (though the authors use a more sophisticated hierarchical Bayesian estimation scheme). Consistent with (some) earlier work, models incorporating both directed exploration (in the form of an exploration bonus tied to bandit arm uncertainty) and perseveration (a tendency to repeat a previous choice) outperform models lacking each of these effects. The main novel finding is that L-dopa reduces the strength of the uncertainty bonus in exploratory choice. While fMRI contrasts between explore and exploit trials did reveal differences in line with previous literature, there were no drug effects that reached statistical significance, save for one exploratory analysis at a reduced threshold. The authors should be commended for comprehensively reporting both positive and null effects. The reviewers also agreed that the paper is for the most part technically well-executed and provides much-needed data on an important set of questions in the reinforcement learning literature.Essential revisions:1) The model comparison includes sensible models (it's also nice to see consistent model ordering across conditions). One thing that's missing (as noted in the Discussion section) is a model of random exploration dependent on "total uncertainty" (see Gershman, 2018, 2019), which should theoretically act as a gain modulator of the values. In fact, what's called the "overall uncertainty" is almost identical to this quantity, and is used in later analyses but apparently not in the choice rule.

As suggested by the Reviewers, we now examined an additional model that includes an additional random exploration term that depends on “total uncertainty” (Gershman et al., 2018), that is, the summed uncertainty across all bandits. This additional model was again combined with both the delta rule (fixed learning rate) and the Kalman filter (uncertainty-dependent learning rate), yielding a total model space of now 8 models.

Based on this expanded model space, we again ran a model comparison via leave-one-out cross validation (Vehtari, Gelman and Gabry, 2017). In short, including an additional term for capturing total uncertainty-based random exploration did not improve model fit compared to the previously best-fitting model with perseveration and directed exploration terms. The revised Figure 3 illustrating the results from the model comparison now reads as follows:

We now describe the results of this expanded model comparison in the Results section, where we now write: “These learning rules were combined with four different choice rules that were all based on a softmax action selection rule (Sutton and Barto, 1998; Daw et al., 2006). Choice rule 1 was a standard softmax with a single inverse temperature parameter (β) modeling random exploration. Choice rule 2 included an additional free parameter (𝜑) modeling an exploration bonus that scaled with the estimated uncertainty of the chosen bandit (directed exploration). Choice rule 3 included an additional free parameter (ρ) modeling a perseveration bonus for the bandit chosen on the previous trial. Finally, choice rule 4 included an additional term to capture random exploration scaling with total uncertainty across all bandits (Gershman, 2018). Leave-one-out (LOO) cross-validation estimates (Vehtari, Gelman and Gabry, 2017) were computed over all drug-conditions, and for each condition separately to assess the models’ predictive accuracies. The Bayesian learning model with terms for directed exploration and perseveration (BayesSMEP) showed highest predictive accuracy in each drug condition and overall (Figure 3). The most complex model including an additional total-uncertainty dependent term provided a slightly inferior account of the data compared to the model without this term (loo log-likelihood: Bayes-SME(R)P: -0.5983 (-0.59989)).”

However, we then had an additional concern regarding our modeling approach. Specifically, we were concerned that our results might be influenced by our model formulation with respect to the implementation of the separate drug sessions. Recall that the original models implemented the three drug sessions via three separate group-level distributions for each parameter (𝛽, 𝜑, 𝜌; mean and standard deviation) from which the single subject parameters for each drug condition were drawn. Alternatively, one could model the placebo condition as the baseline, and the drug-effects as within-subject additive changes from that baseline (see e.g. Pedersen et al., 2017). We therefore re-ran all models and model comparisons using this alternative implementation of the drug-effects.

Specifically, in the alternative model formulation, we modeled potential L-dopa and Haloperidol related deviations (‘shifts’) from the placebo condition in explore/exploit behavior. Separate group-level distributions modeled drug-related deviations from the parameters’ group-level distribution under placebo. On the single subject-level, L-dopa and Haloperidol associated deviations from placebo were then implemented as parameters drawn from these group-level ‘shift*’* distributions (which where modeled with Gaussian priors centered at zero). These ‘shift’ parameters per drug are then added to the placebo parameter estimates via dummy-coded indicator variables coding for the distinct drug conditions (e.g. for the directed exploration parameter 𝜙:

𝜙[𝑠𝑢𝑏𝑗𝑒𝑐𝑡] + 𝐼_𝐻𝐴𝐿_[𝑑𝑟𝑢𝑔] ∗ 𝜙_𝐻𝐴𝐿_[𝑠𝑢𝑏𝑗𝑒𝑐𝑡] + 𝐼_𝐿𝐷_[𝑑𝑟𝑢𝑔] ∗ 𝜙_𝐿𝐷_[𝑠𝑢𝑏𝑗𝑒𝑐𝑡]).

Importantly, using this alternative model formulation, we replicated the model ranking observed for the original formulation. Likewise, the magnitude and directionality of drug effects was highly similar in the alternative model formulation. In Author response image 1, we plot the loo log-likelihood estimates for four models: the best-fitting model (BAYES-SMEP), the model with an additional term for uncertainty-based random exploration (BAYESSMERP), and both models as ‘shift’ versions with drug-effects coded in the manner described above. Therefore, regardless of how drug effects were modeled, the Bayes-SMEP model provided a superior account of the data:

**Author response image 1. respfig1:** Shown are leave-one-out (LOO) log-likelihood estimates calculated for our winning model (BAYES-SMEP), the model with an additional term capturing uncertainty-based random exploration (BAYES-SMERP), and the respective alternative model formulations (‘shift’) over all drug conditions (n=31 subjects with t=3*300 trials) and once separately for each drug condition (n=31 with t=300). All LOO estimates were divided by the total number of data points in the sample (n*t) for better comparability across the different approaches. Bayes: Bayesian learner; SM: softmax (random exploration); E: directed exploration; R: total uncertainty-based random exploration; P: perseveration.

Taken together, both model comparisons suggest that inclusion of an additional term to capture total uncertainty-based random exploration reduces the predictive accuracy of the model. As a consequence, we refrained from examining this model in greater detail with respect to the neuronal and behavioral effects, and rather stuck with the better-fitting original model.

2) It doesn't quite make sense to link the neural representation of "overall uncertainty" to the uncertainty-based exploration strategy formalized in this paper, because that strategy uses an uncertainty bonus, which means that the critical quantity is *relative* not *overall* (or "total" in the terminology of Gershman, 2018) uncertainty (relative and total uncertainty have been distinguished by previous imaging studies: Badre et al., 2012; Tomov et al., 2019). One approach, as suggested above, would be to build uncertainty-based random exploration into the behavioral model, and then connect the quantities in that model to the neural and drug data. Another (not mutually exclusive) approach is to look for neural correlates of relative uncertainty (i.e., the difference between uncertainties). For example, using the uncertainty of the chosen option relative to the average uncertainty of the unchosen options.

We agree that improved visualization of the behavioral effects would be helpful to facilitate understanding the nature of the drug effect. To this end, we now included two additional Figures to visualize the drug effects on behavior.

In the new Figure 2, we now plot the proportion of choices of the best bandit over trials, separately for each drug condition:

With respect to the novel Figure 2 we now write [see Results section]: “Overall, participants’ choice behavior indicated that they understood the task structure, and tracked the most valuable bandit throughout the task. On trial 1, participants randomly selected one of the four bandits (probability to choose best bandit: 21.5% ± 7.49%, M±SE). After 5 trials, participants already selected the most valuable option with 57.76% (±4.89%; M±SE), which was significantly above chance level of 25% (t30=5.79, p=2.52*10-6, Figure 2), and consistently kept choosing the bandit with the highest payoff with on average 67.89% (±2.78%). Thus, participant continuously adjusted their choices to the fluctuating outcomes of the four bandits.”

In the new Figure 5, we now plot the proportion of exploitation, random exploration and directed exploration trials over time, separately for each drug condition:

With respect to the novel Figure 5, we now write [see Results section]: “Next we tested for possible drug effects on the percentage of exploitation and exploration trials (overall, random and directed) per subject. Three separate rmANOVAs with within factors drug and trial (6 blocks of 50 trials each) were computed for each of the following four dependent variables: the percentage of (a) exploitation trials, (b) overall exploration trials, (c) random exploration trials, and (d) directed exploration trials. We found a significant drug effect only for the percentage of directed explorations (F1.66,49.91 =7.18, p = .003; Figure 5c). Fraction of random explorations (F2,60 = 0.55, p = .58, Figure 5b), overall explorations (F2,60 = 0.97, p = .39), or exploitations (F2,60 = 1.57, p = .22; Figure 5a) were not modulated by drug. All drug block interactions were not significant (p>=0.19). Post-hoc, paired ttests showed a significant reduction in the percentage of directed explorations under L-dopa compared to placebo (mean difference P-D = 2.82, t30 = 4.69, p < .001) and haloperidol (mean difference H-D = 2.42, t30 = 2.76, p = .010), but not between placebo and haloperidol (mean difference P-H = 0.39, t30 = 0.43, p = .667). Notably, an exploratory t-test revealed that the percentage of exploitations was marginally increased under L-dopa compared to placebo (mean difference P-D = -2.61, t30 = -1.92, p = .065).”

3) The manuscript is quite long and has a lot of detail that could be relegated to supplemental or summarized. Each subtopic received at least a page in the Discussion, comprising one paragraph of discussion of results followed by lengthy literature review and speculation about its relation to other findings. All of the detail is interesting to a small percentage of expert readers. However, for most readers it makes it hard to get at the main results. As many of the outcomes are null findings, this is even more of an issue. For example, Figure 3 could be removed and replaced with some text indicating means and 95% Cis. Figure 4 is too much detail. Also, the paragraph that discusses how the inclusion of the perseverative coefficient in the model increases sensitivity for directed exploration could be in supplemental (this is worth pointing out, since the original Daw study did not find evidence for this, but this will be of interest to a small number of people, especially since several other studies have now found evidence for an uncertainty bonus). Also, the paragraph that shows no evidence for U-shaped dopamine effects and similarly the figure showing RPE effects in the ventral-striatum could go to supplemental. And correspondingly, a bit more behavioral data might be useful. For Figure 2, is there a way to show a summary figure, across subjects? One of the issues with the original Daw study, was that it was very hard for the participants to learn. How well are the participants able to track the values, and find the best option? Is there a way (might be very hard) to illustrate the effect of L-dopa on directed exploration by directly plotting behavioral data?

We substantially shortened the manuscript and relegated significant parts from the results and Discussion sections to the supplement (‘Appendix 1’). Specifically, the following sections have now been moved to Appendix 1:

1) The delineation of model-based/model-free reinforcement learning and its relation to explore/exploit behavior in the introduction;

2) The description of the assessment of dopamine proxy measures to test dopamine baseline dependent effects within the introduction,

3) The entire section ‘Accounting for perseveration boosts estimates of directed exploration’ within the results;

4) The subsection showing the drug effects on the single-subject parameter posterior distributions within the Results section ‘L-dopa reduces directed exploration’;

5) The section of explore/exploit related brain activation and the respective drug effects has been condensed and the figures depicting the contrasts explore vs exploit, as well as PE-related activation have been moved to Appendix 1.

6) All subsections within the Discussion have been substantially shortened, specifically the sections on drug effects on behavior and neural activation have been significantly condensed.